# Elastic porous microspheres/extracellular matrix hydrogel injectable composites releasing dual bio-factors enable tissue regeneration

Yi Li[1,4], Siyang Liu[1,4], Jingjing Zhang[2,4], Yumeng Wang[1], Hongjiang Lu[1], Yuexi Zhang[3], Guangzhou Song[1], Fanhua Niu[1], Yufan Shen[1], Adam C. Midgley [1], Wen Li[1] ✉, Deling Kong [1] ✉ & Meifeng Zhu [1] ✉

Injectable biomaterials have garnered increasing attention for their potential and beneficial applications in minimally invasive surgical procedures and tissue regeneration. Extracellular matrix (ECM) hydrogels and porous synthetic polymer microspheres can be prepared for injectable administration to achieve in situ tissue regeneration. However, the rapid degradation of ECM hydrogels and the poor injectability and biological inertness of most polymeric microspheres limit their pro-regenerative capabilities. Here, we develop a biomaterial system consisting of elastic porous poly(l-lactide-co-ε-caprolactone) (PLCL) microspheres mixed with ECM hydrogels as injectable composites with interleukin-4 (IL-4) and insulin-like growth factor-1 (IGF-1) dual-release functionality. The developed multifunctional composites have favorable injectability and biocompatibility, and regulate the behavior of macrophages and myogenic cells following injection into muscle tissue. The elicited promotive effects on tissue regeneration are evidenced by enhanced neomusle formation, vascularization, and neuralization at 2-months post-implantation in a male rat model of volumetric muscle loss. Our developed system provides a promising strategy for engineering bioactive injectable composites that demonstrates desirable properties for clinical use and holds translational potential for application as a minimally invasive and pro-regenerative implant material in multiple types of surgical procedures.

Trauma and tumor resection result in tissue defects that require morphological and functional restoration. To promote in situ tissue regeneration, biomaterials have been widely employed as strategies to modulate host cell behavior and stimulate endogenous regeneration potential[1,2]. Biomaterial application or implantation is a simplistic,

effective, and clinically translational approach that avoids limitations associated with traditional tissue engineering[1,3]. Various implantable biomaterials have been developed to improve tissue regeneration, but most of require open surgery that leads to risk of postoperative infection and poor healing outcomes. In contrast, injectable hydrogels

[1]College of Life Sciences, Key Laboratory of Bioactive Materials (Ministry of Education), State Key Laboratory of Medicinal Chemical Biology, Nankai University, Tianjin 300071, China. [2]Chifeng Municipal Hospital, Chifeng 024000 Inner Mongolia, China. [3]The Third Affiliated Hospital of Wenzhou Medical University, Wenzhou 325200 Zhejiang, China. [4]These authors contributed equally: Yi Li, Siyang Liu, Jingjing Zhang. ✉e-mail: liwen1182613400@163.com; kongdeling@nankai.edu.cn; zhumeifeng@nankai.edu.cn

and microspheres provide minimally invasive means of achieving tissue regeneration[4,5] coupled with the advantages of minor tissue injury, fast recovery, convenience of operation, and clinical safety. However, when injected, microspheres or hydrogels suffer from unavoidable drawbacks, such as deformation and rapid degradation, respectively[5,6]. Therefore, in recent years there has been considerable efforts made to develop functional injectable biomaterials that promote in situ tissue regeneration[7,8].

Extracellular matrix (ECM) hydrogels and synthetic polymer microspheres are commonly used as injectable biomaterials for the repair of injured tissues[4,9–18], including cartilage[9], intervertebral discs[10,16], muscle[11], and bone[17,18], among others[12–15]. Numerous studies have indicated that synthetic polymer microspheres with controllable pore structures and high porosity can serve as vehicles for the loading of live cells or as factor-eluting beacons to recruit endogenous cells[19,20]. However, most porous microspheres (PM) prepared from synthetic polymers lack elasticity, and thus easily undergo deformation under compression and wall shear stress during injection, which can result in the structural and morphological collapse of microspheres. In addition, synthetic polymers lack bioactivity[5,21], and typically, their degradation leads to a buildup of acidic byproducts that limit pro-regenerative effects[22]. ECM hydrogel components closely mimic the ECM arrangement of native tissues whilst also possessing high biocompatibility and temperature-responsive gel formation properties[6,23]. However, ECM hydrogels exhibit weak mechanical strength and rapid degradation in vivo, which presents a challenge in providing the necessary mechanophysical support for cell and tissue growth[6]. Therefore, constructing injectable composites that combine the complementary advantages gained from elastic synthetic polymer microspheres and ECM hydrogels can offer a biomaterial system that effectively promotes tissue regeneration.

Biomaterial-induced in situ tissue regeneration is a sequential biological process involving multiple cells[1,24,25]. First, monocytes are recruited to the injury site by neutrophils and transform into macrophages, which triggers a tissue repair response by releasing cytokines. Subsequently, endogenous cells including tissue-resident stem cells and somatic cells, as well as progenitors recruited from the bloodstream, migrate, proliferate, and differentiate, whilst secreting new ECM to form vascular and neural networks at the injury site. Finally, the injured tissue gains restored physiological function through ECM degradation, resynthesis, and remodeling. Therefore, the simultaneous regulation of macrophage and endogenous stem cell behavior may serve as a novel strategy to enhance in situ tissue regeneration. Interleukin 4 (IL-4) has verified roles in promoting tissue repair via the polarization of macrophages[26–33], including in bone[28,31], vascular tissues[29,30], and during the repair of ischemic stroke injury[32]. Additionally, insulin-like growth factor (IGF-1) has been shown to promote stem/progenitor cell proliferation, differentiation, and maturation[34–38], in muscle[35,39], peripheral nerve[36], and cartilage tissue regeneration[37,40]. Therefore, we anticipate that the controlled release of IL-4 and IGF-1 from injectable composite biomaterials can facilitate the simultaneous modulation of macrophage and stem cell behavior, to enhance in situ tissue regeneration.

Here, based on the favorable properties identified in our previous study[41], we select elastic biodegradable poly(l-lactide-caprolactone) (PLCL) to prepare PM. The elastic PLCL PMs are synthesized using a combination of microfluidic and polymer leaching technology, before further modification with polydopamine (PDA)-conjugated IGF-1 (PM@IGF1) (Fig. 1A). IL-4 is physically loaded into the porcine muscle-derived ECM (mECM) hydrogel (mECM@IL4). Then, PM@IGF1 microspheres and mECM@IL4 hydrogels are mixed in optimum proportions to achieve bioactive injectable composites (mECM@IL4 + PM@IGF1) (Fig. 1B). We characterize the structure, mechanical, and physical properties of the elastic PM, mECM hydrogels, and composites. We also investigate the release behavior of mECM@IL4 + PM@IGF1 and its regulatory effect on macrophages and myogenic

cells in vitro. Finally, we demonstrate the injectability, biocompatibility, and pro-regenerative effects of mECM@IL4 + PM@IGF1 composites by subcutaneous injection and in rat models of volumetric tibialis anterior muscle loss (Fig. 1C).

## Results

### Preparation of elastic porous microspheres, polydopamine modification and characterization

Three kinds of synthetic polymers, PLCL, poly (lactic acid/glycolic acid) (PLGA), and poly(caprolactone) (PCL), were used to prepare PM using a combination of microfluidics and gelatin leaching methods. Scanning electron microscopy (SEM) indicated that PLCL, PLGA, and PCL microspheres maintained stereoscopic spherical structures and possessed highly interconnected pore structures at the surfaces and throughout the interiors (Fig. 2A). The diameters of the PLCL, PLGA, and PCL microspheres were $345.7 \pm 42.8\,\mu m$, $383.5 \pm 66.6\,\mu m$, and $392.9 \pm 65.2\,\mu m$, respectively (Fig. 2B). The diameters of the pores on the surface and cross-section of the PLCL microspheres were $48.9 \pm 14.3\,\mu m$ and $39.3 \pm 13.8\,\mu m$, without significant statistical differences (Fig. 2C). The surface pore diameters of PLGA and PCL microspheres were $29.7 \pm 7.6\,\mu m$ and $45.7 \pm 9.8\,\mu m$, respectively (Fig. 2C). Furthermore, polydopamine (PDA) modification did not affect the diameter and pore size of the microspheres (Fig. 2A-C). The average diameter of PDA-modified PLCL microspheres (PLCL@PDA; PM) was $355.3 \pm 37.4\,\mu m$, and the average pore sizes on the surface and cross-section were $45.7 \pm 13.3\,\mu m$ and $41.3 \pm 12.2\,\mu m$, respectively (Fig. 2B, C). In addition, we demonstrated the accurate control of microsphere diameters (300–800 μm) and pore sizes (20–100 μm) by regulating the microfluidic needle size and the gelatin concentration used in polymer leaching, respectively (Supplementary Fig. 1, 2). Atomic force microscopy (AFM) showed that the average roughness (RA) of the PLCL microsphere surface was significantly improved after PDA modification, which was confirmed by statistical analysis (Fig. 2D, E). Additionally, X-ray photoelectron spectroscopy (XPS) showed that the peak value of $N\,1s$ had increased, verifying successful PDA modification (Fig. 2F). We then tested the injectability and mechanical properties of the different microspheres. After syringe injection, PLCL microspheres maintained their original morphology. In contrast, PLGA and PCL microspheres frequently exhibited breaks and deformations (Fig. 2G). Moreover, PLCL microspheres recovered to $88.7 \pm 4.8\%$ and $90.5 \pm 4.0\%$ of their original size after compression with stainless steel tweezers for 30 s and 2 min, respectively. The same tests resulted in the PLGA microspheres being crushed and the PCL microspheres flattened, without the ability to recover original sizes (Fig. 2H, Supplementary Fig. 3). Similarly, stereomicroscopy and captured movies showed that PLCL microspheres maintained their original size after being pressed for 15 times with tweezers, whereas the PLGA and PCL microspheres were crushed or deformed after only 1–3 presses (Fig. 2I, J, Supplementary movies 1–3). The morphology recovery time of compressed PLCL microspheres increased as their diameter and porosity enhanced (Tables 1, 2). Finally, in light of their elasticity and resilience, PLCL microspheres were selected for use in subsequent experiments.

### Preparation and characterization of mECM hydrogels and mECM + PM composites, and the dual-release profiles of mECM@IL4 + PM@IGF1

Next, we prepared porcine mECM hydrogels by combining physical, chemical, and biological processing methods (Fig. 3A). Muscle tissue turned white and translucent after decellularization. H&E and Masson's trichrome staining indicated the retention of ECM, muscle fibers, and a small amount of collagen (Fig. 3B). Following decellularization, DAPI staining and corresponding statistics showed that $97.9 \pm 2.0\%$ of the cell nuclei were removed. The DNA content decreased from $536.2 \pm 70.4\,ng/mg$ to $37.5 \pm 2.2\,ng/mg$, as detected by a DNA

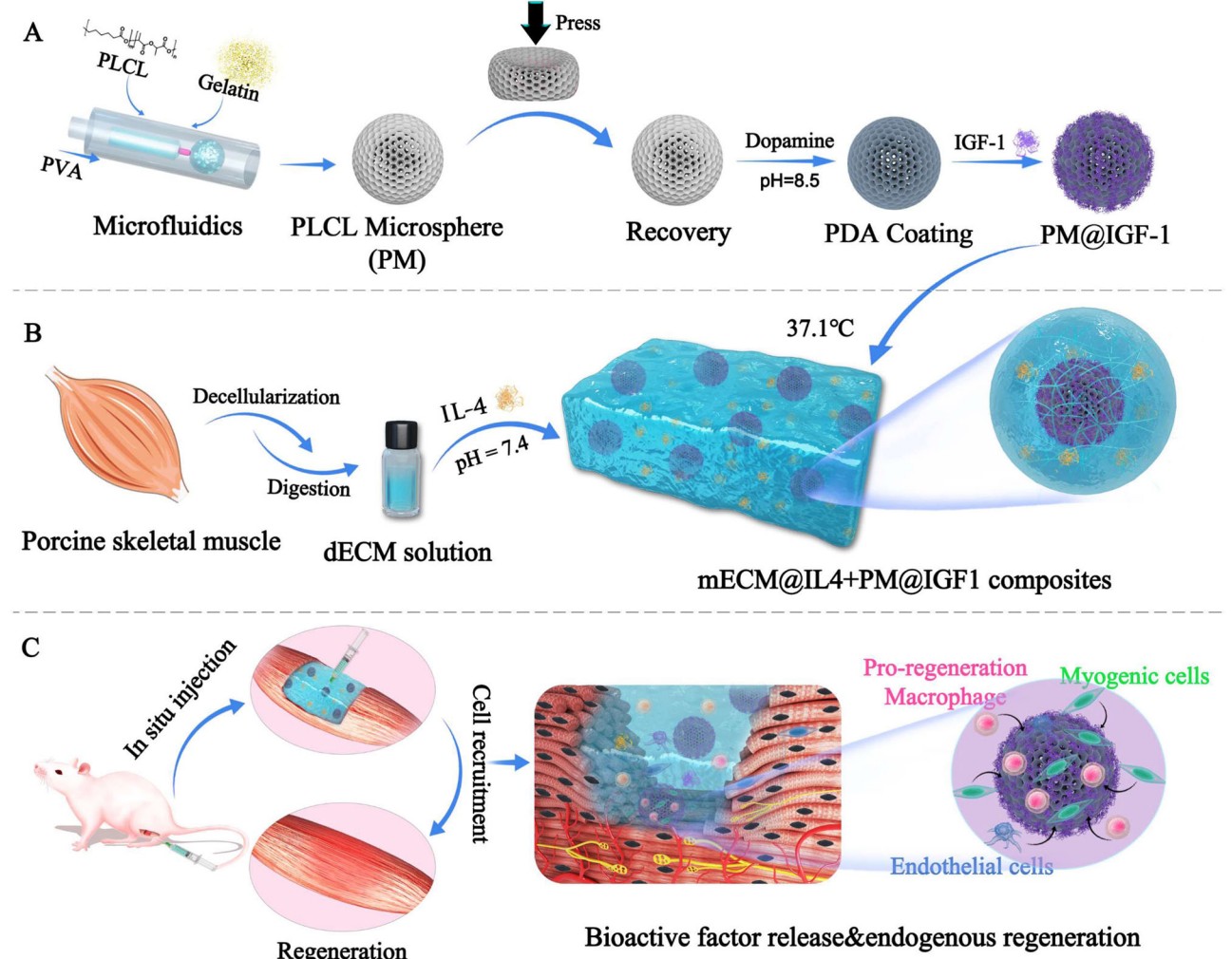

**Fig. 1 | Sketch map of the preparation and application of mECM@IL4 + PM@IGF1 composites. A** Fabrication of elastic porous PLCL microspheres modified with PDA-linked IGF-1. **B** Preparation of mECM@IL4 + PM@IGF1 composites. **C** In situ injection of composites to promote endogenous muscle regeneration.

quantitative kit (Fig. 3B, F, G), which was lower than the 50 ng/mg content threshold that indicates sufficient decellularization[42]. The activity of α-gal also decreased from 2689.9 ± 202.7 U/g to 287.0 ± 148.9 U/g, as tested by α-galactosidase quantitative kit (Fig. 3H). We next screened the gel-forming capabilities and conditions of mECM. The 1% mECM hydrogel was unable to form a hydrogel and the 2% mECM hydrogel was unstable. The 3% mECM formed a stable hydrogel exhibiting a typical gel-like appearance and shape, which could be physically handled and picked up with tweezers without breakage (Fig. 3C, Supplementary Fig. 4). Thus, we selected 3% mECM for constructing composites. SEM showed that the surface and cross-section of the 3% mECM hydrogel had a uniform micropore structure, with surface and interior pore sizes of 33.9 ± 11.5 μm and 63.4 ± 23.2 μm, respectively (Fig. 3D). To screen the optimal concentration of composites, we investigated the effect of different mECM to PM mass ratio on the injectability, stability, mechanics and pro-regenerative effects of composites (Supplementary movies 4−7; Supplementary Figs. 5, 6), the mass ratio of 10:1 was selected for use in subsequent experiments. Under dry conditions, the microspheres were distributed throughout the mECM hydrogel, and mECM was observed to also be anchored within the porous walls of the PLCL microspheres, as visualized by SEM (Fig. 3D). Under wet conditions, rhodamine-labeled PLCL microspheres and FITC-labeled mECM hydrogels formed uniform and interpenetrating composite structures (Fig. 3E). Rheology testing indicated that the mECM solution was

unable to form a hydrogel, and microspheres provided an initial storage modulus ($G'$) of ~40 Pa, when the temperature was below 37 °C (Fig. 3I, J, L). After gelling at 37.1 °C, the storage modulus ($G'$) and loss modulus ($G''$) of the mECM hydrogel and composites were both enhanced, but $G'$ exhibited more obvious enhancements (Fig. I–L). Supplementary Figs. 5B–E and supplementary Figs. 7 showed that the lower density and higher porosity of microspheres resulted in higher mechanical strength of composites. Afterwards, IGF-1 was modified onto the porous PLCL microspheres via PDA coatings (PM@IGF1), and it exhibited a similar release behavior with or without additional mechanical actions (injection and compression) (Supplementary Fig. 8). Then PM@IGF-1 was mixed with IL-4-loaded mECM hydrogels (mECM@IL4) to yield composites (mECM@IL4 + PM@IGF1). The cytokine/growth factor dual-release behavior was also detected by ELISA. The release curve of IL-4 from the composites was similar to that of the 3% mECM hydrogel, and it showed a rapid release within 3 days (Fig. 3M). The release speed of IGF-1 from the mECM@IL4 + PM@IGF-1 composites was slower than that measured from PM@IGF1 alone, demonstrating an approach to complete release by day 10 (Fig. 3N).

## The mECM@IL4 + PM@IGF1 composites cooperatively induce the polarization of primary macrophages into pro-regenerative phenotypes

We used transwell coculture assays to evaluate the regulatory effect of the composite on the behavior of primary bone marrow-derived

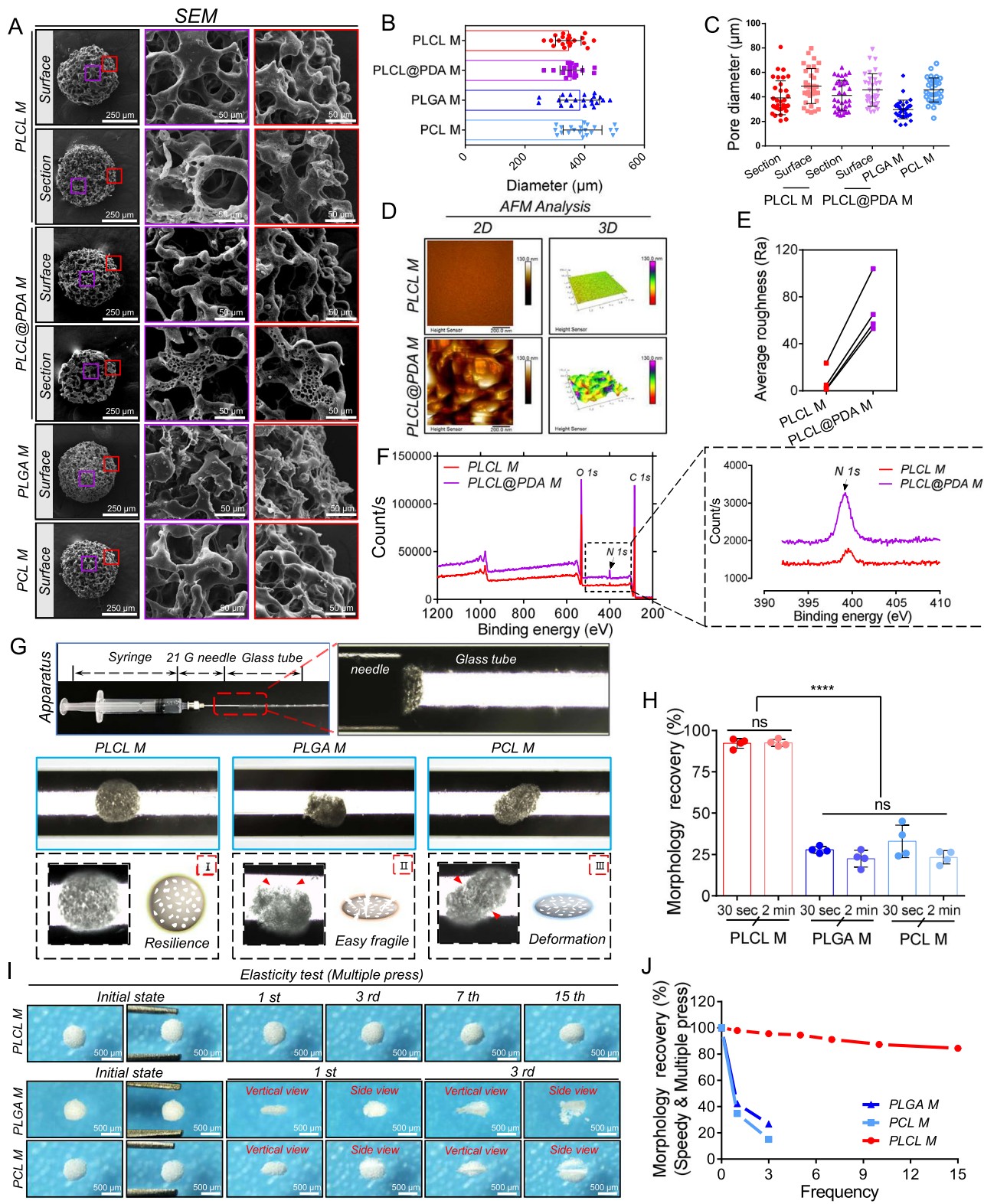

macrophages (BMDMs) (Fig. 4A). CD68 and CD206 immuno-fluorescence staining showed that all IL-4 groups (mECM@IL4 + PM, mECM@IL4 + PM@IGF1, and pure IL-4 group) significantly increased the expression of CD206 on day 1 (Fig. 4B). Although the expression of CD206 tended to be higher in the IL-4 group than in the mEC-M@IL4 + PM and mECM@IL4 + PM@IGF1 groups, no significant differences were observed (Fig. 4C). The expression of CD206 in the IL-4 group showed no significant change from day 1 to day 3, but increased in the other groups (Fig. 4B, C). On day 3, the expression of CD206 in

the mECM@IL4 + PM@IGF1 group was higher than that in the other groups, and showed a significant difference compared with the control, mECM+PM, and mECM + PM@IGF1 groups (Fig. 4B, C). The pro-regenerative macrophages mainly showed elongated cell morphology, while the pro-inflammatory macrophages were mostly rounded in appearance (Fig. 4D). The elongation values of BMDMs in the IL-4 groups (mECM@IL4 + PM, mECM@IL4 + PM@IGF1, and IL-4) tended to be higher than those in groups without IL-4 on day 1, despite displaying no significant differences. However, the elongation value in the

**Fig. 2 | Preparation, polydopamine modification, and mechanophysical characterization of porous synthetic polymer microspheres. A** SEM images showing the macro-morphology and pore structure of different microspheres. Purple box: central region; red box: marginal region. **B** Respective statistical results of different microspheres' diameter. Data are presented as mean ± SD ($n = 21$ independent microspheres, from three independent batches of samples). **C** Statistical results of different microspheres' pore sizes. Data are presented as mean ± SD ($n = 34$ independent pore sizes, from three independent batches of samples). **D, E** 2D and 3D images (AFM) showing the surface roughness of porous PLCL microspheres before and after PDA modification, as well as corresponding statistical analysis. Data are presented as mean ± SD ($n = 4$ independent samples). **F** XPS analysis of C, O, and N peak changes before and after PDA modification, dotted box: local enlargement of the *N 1s* peak. **G** The injectability of different microspheres. I–III: Morphology pattern diagrams of different microspheres after injection with a syringe. Red box: high magnification images; Red arrow: microsphere fragmentation/deformation area. **H** Recovery rate of different microspheres after compressing once for 30 s or 2 min. Data are presented as mean ± SD ($n = 4$ independent samples). ****$P < 0.0001$, ns: no significant difference, one-way ANOVA, multiple comparisons. **I, J** The morphology of different microspheres after repeated compressions and the corresponding recovery rate statistics.

### Table 1 | Recovery time of porous PLCL microspheres with different needle sizes

| Different needle sizes | Diameter (μm) (Mean ± SD) | Recovery time (s) (Mean ± SD) |
|---|---|---|
| 18 G | 818.1 ± 88.3 | 2.13 ± 0.22 |
| 20 G | 607.3 ± 73.7 | 1.91 ± 0.16 |
| 21 G | 544.4 ± 42.5 | 1.78 ± 0.19 |
| 26 G | 360.0 ± 28.6 | 1.51 ± 0.27 |

### Table 2 | Recovery time of porous PLCL microspheres with different gelatin concentrations

| Different gelatin concentrations (%) | Porosity (%) (Mean ± SD) | Recovery time (s) (Mean ± SD) |
|---|---|---|
| 2.5 | 43.5 ± 7.5 | 1.07 ± 0.23 |
| 5.0 | 76.6 ± 5.1 | 1.45 ± 0.40 |
| 7.5 | 83.7 ± 2.8 | 1.42 ± 0.42 |
| 10.0 | 92.8 ± 1.6 | 2.89 ± 0.65 |

mECM@IL4 + PM and mECM@IL4 + PM@IGF1 groups exhibited no significant differences compared with the pure IL-4 group (Fig. 4E). The BMDMs elongation value of the IL-4 containing groups on day 3 showed an increasing trend compared with the values measured on day 1. The BMDMs elongation values in the mECM@IL4 + PM and mECM@IL4 + PM@IGF1 groups were higher than those in the pure IL-4 group on day 3, although again, this showed no significant differences (Fig. 4E).

We next detected the proportion of CD206$^+$ cells by flow cytometry. On day 1, the proportion of CD206$^+$ cells increased slightly in the mECM + PM and mECM + PM@IGF1 groups compared with the control (Supplementary Fig. 9A-C). The proportion of CD206$^+$ cells in the mECM@IL4 + PM and mECM@IL4 + PM@IGF1 groups significantly increased to 20.2 ± 1.5% and 20.0 ± 1.3%, respectively, and both showed no significantly differences compared with the pure IL-4 group (Supplementary Fig. 9C). Additionally, the proportion of CD206$^+$ cells were significantly increased in all groups from day 1 to day 3, except the pure IL-4 group (Fig. 4F-H, Supplementary Fig. 9D). On day 3, the proportion of CD206$^+$ cells in the mECM + PM group was still slightly increased compared with the control group (17.5 ± 0.8% vs. 14.9 ± 0.9%), but exhibited no significantly differences (Fig. 4F, H). The proportions of CD206$^+$ cells in the mECM + PM@IGF1 and mECM@IL4 + PM groups were significantly increased compared with that in the control and mECM + PM group. Notably, the proportion of CD206$^+$ cells in the mECM@IL4 + PM@IGF1 group was considerably greater when compared with the mECM + PM@IGF1 and mECM@IL4 + PM groups (Fig. 4F, H). In addition, there was no significant change in CD206$^+$ cell proportion in the pure IL-4 group on day 3 and day 1 (Supplementary Fig. 9D). Indeed, the number of CD206$^+$ cells in the pure IL-4 group was significantly lower than that in the mECM@IL4 + PM and mECM@IL4 + PM@IGF1 groups on day 3 (Fig. 4F), which was due to slower release of IL-4 encapsulated in mECM hydrogel. The CD206 peak value shift and corresponding statistical analysis also supported the abovementioned measurements (Fig. 4G, H, Supplementary Fig. 9B, C).

We then detected BMDMs secretion of inflammation-related factors by Luminex liquid chip; the results on day 1 and days 3 were respectively showed in Fig. 4I-N and Supplementary Fig. 10. On day 1, the pro-inflammatory factors IL-1β, IL-6, IL-12P70 and MCP-1, IL-1α, interferon (IFN)-γ and tumor necrosis factor (TNF)-α in the IL-4 containing groups (mECM@IL4 + PM, mECM@IL4 + PM@IGF1, and pure IL-4 groups) exhibited a downward trend, but there was no significant difference among the three groups (Fig. 4I-L, Supplementary Fig. 10A–C). IL-5 and GM-CSF displayed no significant changes among all groups (Supplementary Fig. 10D, E). The secretion of anti-inflammatory factors, including IL-4 and IL-10, in the mECM@IL4 + PM, mECM@IL4 + PM@IGF1, and pure IL-4 groups was higher than that in the groups without IL-4 (Fig. 4M, N). The secretion of IL-4 only in pure IL-4 groups was significantly higher than that in the mECM + PM group (Fig. 4M). While the secretion of IL-10 in the mECM@IL4 + PM, mECM@IL4 + PM@IGF1, and pure IL-4 groups was significantly higher than that in groups without IL-4, and there were no significant differences among the three groups (Fig. 4N). On day 3, the secretion of IL-1β, IL-6, IL-12P70, MCP-1, IL-1α, IFN-γ, TNF-α, and IL-5 in the mECM@IL4 + PM@IGF1 group was lower than that in the other groups (Fig. 4I-L, Supplementary Fig. 10A-D). The secretion of GM-CSF still displayed no significant changes among all groups (Supplementary Fig. 10E). Furthermore, IL-1β, IL-6, IL-12P70, and MCP-1 in the mECM@IL4 + PM@IGF1 group displayed significant differences compared with control and mECM + PM groups (Fig. 4I, J, K, L). The expression of IL-1α in the mECM@IL4 + PM@IGF1 group was significantly lower than that in the control, mECM + PM and mECM + PM@IGF1 groups (Supplementary Fig. 10A). Moreover, the expression of IFN-γ, TNF-α, and IL-5 in the mECM@IL4 + PM and mECM@IL4 + PM@IGF1 groups was significantly lower than that in the control group (Supplementary Fig. 10B–D). However, the secretion of IL-4 in the mECM@IL4 + PM@IGF1 group was consistently higher than that in the other groups, although there was no significant difference compared with the mECM@IL4 + PM group (Fig. 4M). IL-10 in the mECM@IL4 + PM@IGF1 group was significantly higher than that in all other groups (Fig. 4N).

### The regulatory effect of BMDMs and composites on the behavior of L6 cells and injured primary satellite cells

We firstly evaluated the effects of IL-4 and IGF-1 on the viability of L6 cells using CCK-8 arrays, and IGF-1 promoted L6 cell viability while IL-4 displayed no obvious effects (Supplementary Fig. 11). Then, we used transwell coculture models to evaluate the regulatory effects of BMDMs and composites on the behaviors of L6 myoblast cells and injured primary satellite cells. Live/dead staining showed that L6 cells in all groups retained high viability and exhibited spreading morphologies on day 3 (Supplementary Fig. 12A). On day 5, cells still maintained high viability in all groups, and cell densities in the

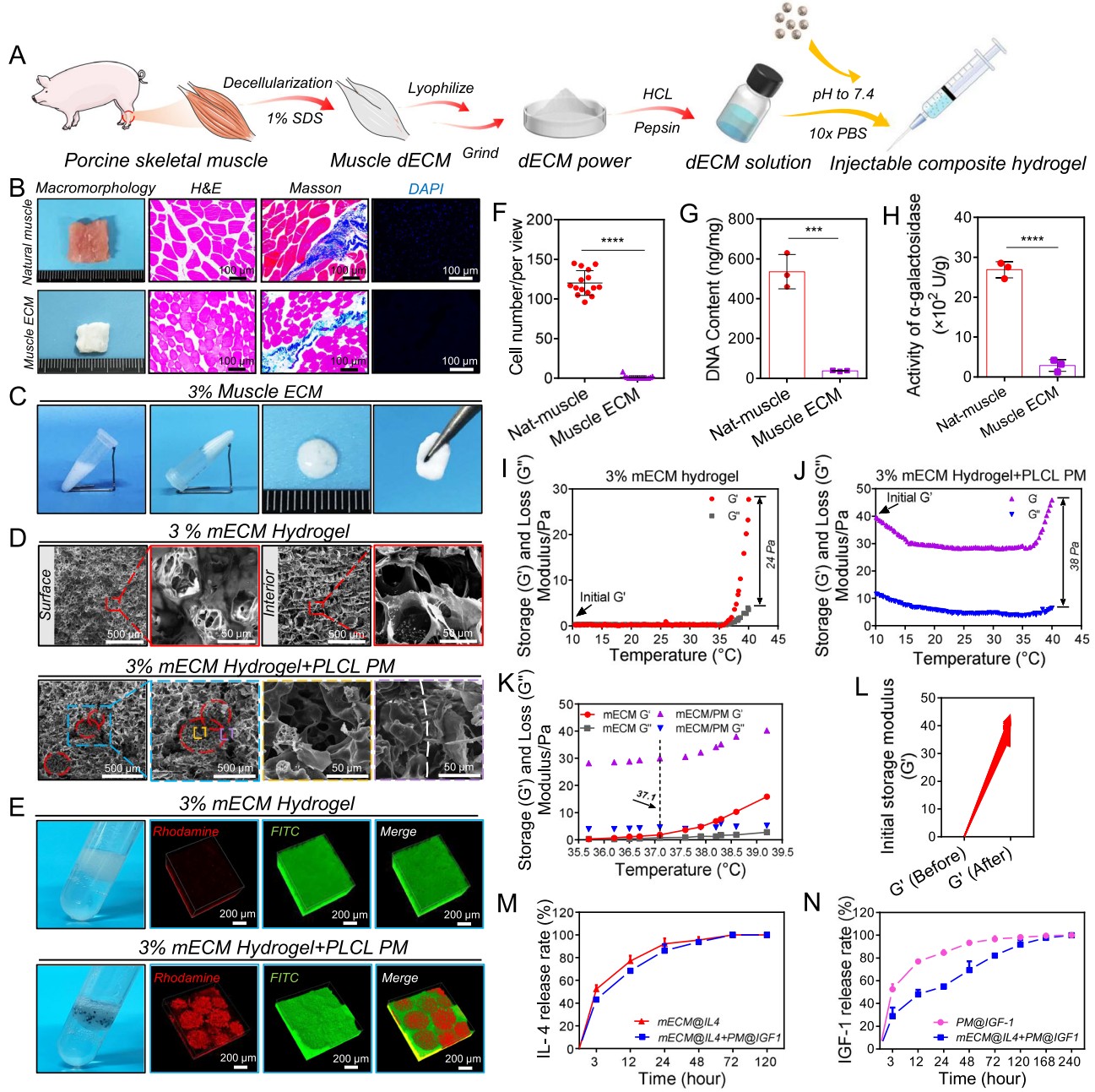

**Fig. 3 | Preparation and characterization of the mECM hydrogels, mECM + PM composites, and the release behavior of mECM@IL4 + PM@IGF1. A** Schematic diagram of mECM + PM composite preparation. **B** Stereomicroscopy, H&E, Masson's trichrome, and DAPI staining of porcine skeletal muscle before and after decellularization. **C** Gel-forming properties of the 3% mECM hydrogel. **D** SEM images showing the structure and composition on the surfaces and cross-sections of 3% mECM and composite. Red and light blue dashed box: high magnification images; red dashed circle: microspheres; orange dashed box: central region of microsphere; purple dashed box: junction region of microspheres and mECM; white dashed line: junction of microspheres and mECM. **E** Stereomicroscopic images of the mECM hydrogel and composite, and fluorescent 3D images of rhodamine-labeled microspheres (red) and FITC-labeled mECM (green)

composites. **F** Quantitative analysis of DAPI staining in Fig. 3B. Data are presented as mean ± SD (*n* = 3 independent samples, 5 random fields per sample). ****$P$ < 0.0001, *t* test, two-tailed. **G** DNA quantitative analysis before and after decellularization. Data are presented as mean ± SD (*n* = 3 independent samples). ***$P$ < 0.001, *t* test, two-tailed. **H** Detection of α-galactosidase before and after decellularization. Data are presented as mean ± SD (*n* = 3 independent samples). ****$P$ < 0.0001, *t* test, two-tailed. **I, J** Rheological tests of the 3% mECM hydrogel and composite. **K** Gel-forming property. **L** The initial elastic modulus (*G'*) of 3% mECM hydrogels and the mECM + PM composites. Data are presented as mean ± SD (*n* = 3 independent samples). **M, N** Release profiles of IL-4 and IGF-1 in the mECM@IL4 + PM@IGF1 composites. Data are presented as mean ± SD (*n* = 2 independent samples). For (**D, E**), the experiment was repeated three times independently.

IGF-1-loaded groups (PM@IGF1, mECM + PM@IGF1, and mECM@IL4 + PM@IGF1) were higher than that in the groups without IGF-1 (Supplementary Fig. 12A). On day 7, the cell survivability and densities in the IGF-1-loaded groups (PM@IGF1, mECM + PM@IGF1, and mECM@IL4 + PM@IGF1 groups) were evidently higher than those in the

control, mECM+PM, and pure IGF-1 groups (Supplementary Fig. 12A). Furthermore, the number of dead cells exhibited no significant differences among all groups on days 3 and 5 (Supplementary Fig. 12B). On day 7, the number of dead cells in the IGF-1-loaded groups (PM@IGF1, mECM + PM@IGF1, and mECM@IL4 + PM@IGF1 groups)

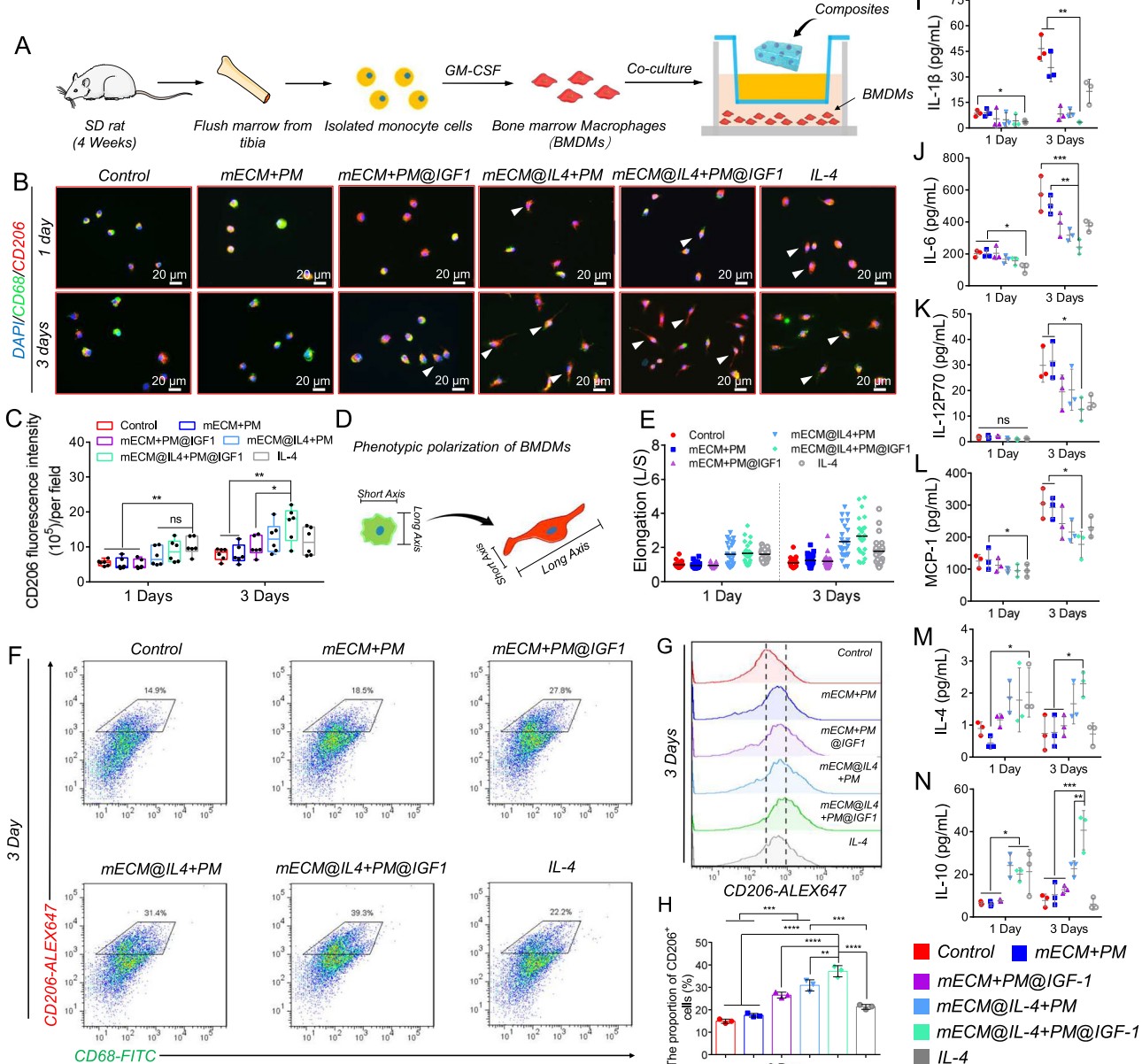

**Fig. 4 | The mECM@IL4 + PM@IGF1 composites induce the polarization of the pro-regenerative macrophage phenotype. A** Diagram of bone marrow-derived macrophages (BMDMs) extraction and coculture with composites. **B** Fluorescence images of CD68 (green) and CD206 (red) after different treatments. White arrows: morphologically extended macrophages; nuclei: blue. **C** Corresponding statistical analysis of CD206 fluorescence intensity. The error bars represent the minimum and maximum values; the line inside the box represents the median; box limits represent the 25th and 75th percentiles (*n* = 6 biologically independent samples). **P < 0.01, *P < 0.05, ns: no significant difference, one-way ANOVA, multiple comparisons. **D, E** Schematic diagram of the morphological change of BMDMs and corresponding statistics. Data are presented as mean ± SD (*n* = 25 biologically independent cells). **F** Flow cytometric analysis of BMDMs showing the expression of CD206⁺ gated cells after different treatments for 3 days. **G, H** Histogram of the peak shift of CD206 expression and corresponding statistical results. Data are presented as mean ± SD (*n* = 3 biologically independent samples). ****P < 0.0001, ***P < 0.001, **P < 0.01, one-way ANOVA, multiple comparisons. **I–N** Luminex microarrays detecting the inflammation-related factors secreted by BMDMs after different treatments on day 1 and day 3. Data are presented as mean ± SD (*n* = 3 biologically independent samples). ***P < 0.001, **P < 0.01, *P < 0.05, ns: no significant difference, one-way ANOVA, multiple comparisons.

was significantly lower than that in the unloaded groups and pure IGF-1 group (Supplementary Fig. 12B).

TUNEL staining was used to evaluate the anti-apoptotic effect of the composites on L6 cells. The number of TUNEL-positive cells in the mECM@IL4 + PM@IGF1 group was lower than that in other groups on day 7, indicating significant differences compared with the control, mECM + PM, and pure IGF-1 groups (Fig. 5A, B). CCK8 assays were used to detect L6 cell viability after different treatments (Fig. 5C). On day 1, there were no significant differences among all the groups. On day 3, cell viability in the mECM@IL4 + PM@IGF1 group was higher than that

in control, mECM + PM, PM@IGF1, and mECM + PM@IGF1 groups and lower than that in the pure IGF-1 group, but without a statistical difference. On days 5 and 7, cell viability in the mECM@IL4 + PM@IGF1 group was significantly higher than that in the control, mECM + PM, PM@IGF1 and pure IGF-1 groups, and displayed no significant difference compared with the mECM + PM@IGF1 group (Fig. 5C).

Subsequently, we constructed an in vitro CTX-induced cell injury model to investigate the regulatory effect of BMDMs and composites on the behavior of injured primary satellite cells (Fig. 5D). The intracellular Ca²⁺ fluorescence intensity of satellite cells increased by

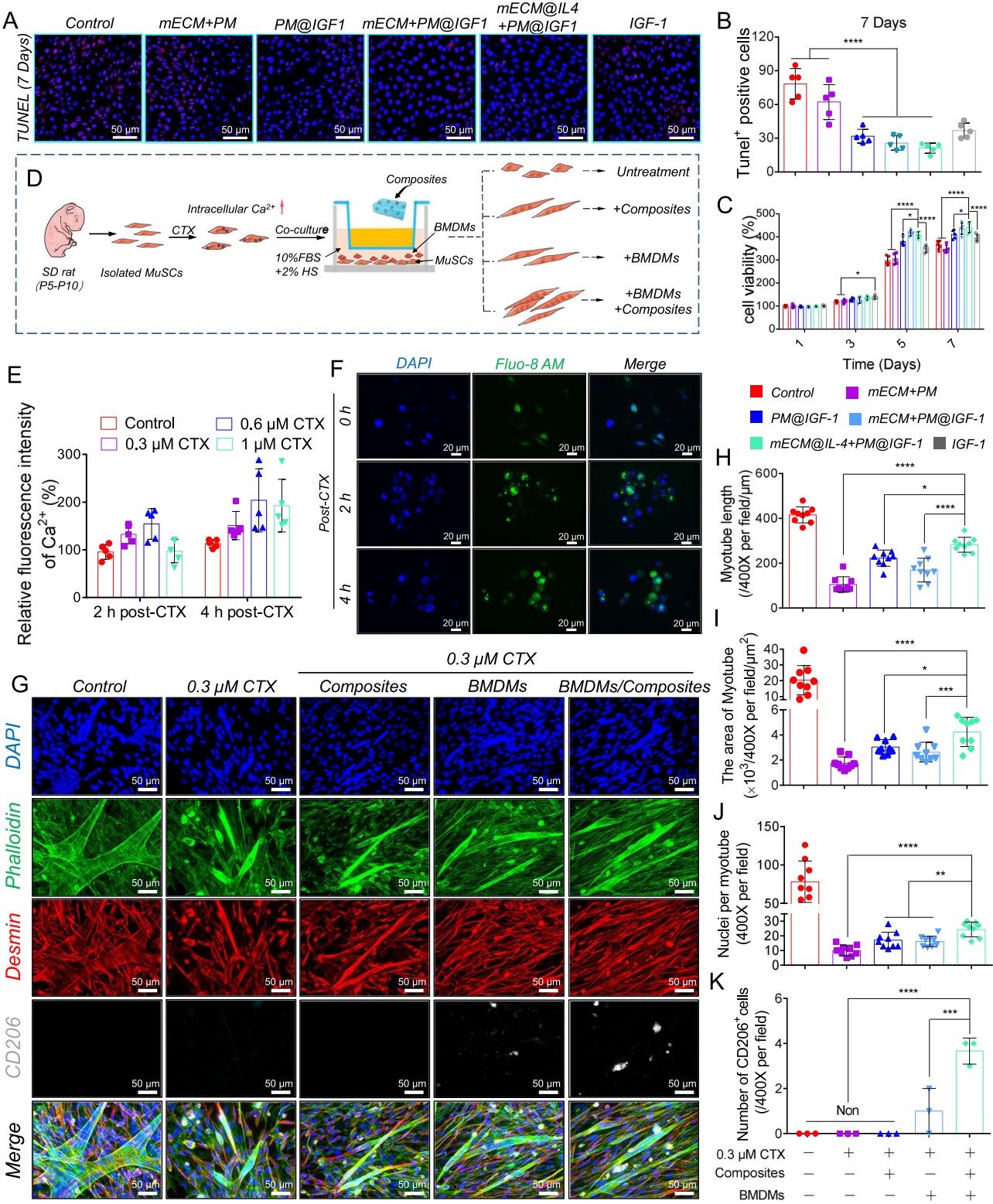

30-40% after treatment with 0.3 μM CTX for $2.0 \pm 0.5$ h, indicating successful construction of the cell injury model (Fig. 5E, F). Then, we used transwell coculture assays to investigate the effect of BMDMs and composites on the differentiation behavior of injured muscle satellite cells (Fig. 5G-K). The uninjured satellite cells in the control group could differentiate into myotubes normally and fuse extensively. In contrast, satellite cells exposed to CTX could be hard to differentiate into myotubes and showed a random distribution and arrangement. The mECM@IL4 + PM@IGF1 composites cultured group (CTX +

composites), the BMDMs cultured group (CTX + BMDMs), and the BMDMs cooperated with mECM@IL4 + PM@IGF1 composites group (CTX + BMDMs/composites) all promoted myotube formation. Moreover, the CTX + BMDMs/composites group exhibited the strongest promotive effects on myotube formation (Fig. 5G). The myotube length and area in the CTX + BMDMs/composites group was significantly higher than that in the CTX group, the CTX + composites group, and the CTX + BMDMs group (Fig. 5H, I). In addition, the number of myonuclei in a single myotube in the CTX + BMDMs/

**Fig. 5 | mECM@IL4 + PM@IGF1 composites modulated the behavior of myogenic cells. A** TUNEL staining of L6 cells on day 7. Nuclei: blue; apoptotic cell: red. **B** Statistical results of the number of TUNEL$^+$ apoptotic cells on day 7. Data are presented as mean ± SD ($n = 5$ biologically independent samples). ****$P < 0.0001$, one-way ANOVA, multiple comparisons. **C** Cell viability of L6 cells treated with different extracting solutions on days 1, 3, 5, and 7. Data are presented as mean ± SD ($n = 5$ biologically independent samples). ****$P < 0.0001$, *$P < 0.05$, one-way ANOVA, multiple comparisons. **D** Schematic diagram showing the regulatory effects of BMDMs and composites on injured primary muscle satellite cells. **E** Ca$^{2+}$ fluorescence intensity (490 nm) of myocytes treated with different CTX concentrations (0, 0.3 μM, 0.6 μM, and 1 μM) for 2 and 4 h. Data are presented as mean ± SD ($n = 5$

biologically independent samples). **F** Fluo-8 AM probe detecting the intracellular Ca$^{2+}$ changes (green) after treatment with 0.3 μM CTX for 2 and 4 h. Nuclei: blue. **G** Representative images of phalloidin (green) co-staining with CD206 (gray) and desmin (red) antibody showing the differentiation-promoting effect of BMDMs and composites on injured muscle satellite cells. Nuclei: blue. **H–J** Quantitative statistics of the myotube length, the myotube fusion area, and the nuclei number per myotube. Data are presented as mean ± SD ($n = 9$ biologically independent myotubes). **K** Quantitative statistics of CD206 fluorescence intensity. Data are presented as mean ± SD ($n = 3$ biologically independent samples). ****$P < 0.0001$, ***$P < 0.001$, **$P < 0.01$, *$P < 0.05$, one-way ANOVA, multiple comparisons.

composites group was also significantly higher than that in the CTX group, the CTX + composites group, and the CTX + BMDMs group, although it was still lower than the control group (Fig. 5J). Furthermore, the composites could also induce macrophage polarization, the number of CD206$^+$ cells in the CTX + BMDMs/composites group was significantly higher than that in CTX + BMDMs group. (Fig. 5G, K).

**Composites exhibited high injectability and histocompatibility**
We verified the injectable properties of the composites using a rat subcutaneous injection model (Fig. 6A). The mECM hydrogel was milky white in the syringe without signs of deposition or separation. After loading into the syringe, the microspheres aggregated and settled towards the bottom in the PM group. However, the microspheres were dispersed uniformly when mixed in with the mECM hydrogel and did not exhibit signs of sedimentation in the mECM + PM group (Fig. 6B). During the injection, microspheres alone were prone to clogging the needle in the PM group, but both the mECM hydrogel and mECM + PM composites could be injected smoothly without disruption to the flow (Supplementary movies 8-10). After injection, the mECM hydrogel, porous PLCL microspheres, and mECM + PM composites were observable after 1 week. With prolonged implantation time, the mECM hydrogel completely degraded, whereas microspheres could still be detected in the PM and mECM + PM groups at week 4 (Fig. 6B).

H&E staining showed that the host cells were mostly distributed around the periphery of materials in the mECM and mECM + PM groups, while cells had quickly infiltrated into the interiors of the microspheres in the PM group at week 1 (Fig. 6C). The microspheres in the PM group displayed heterogeneous degradation and collapse after 4 weeks of implantation, whereas the microspheres in the mECM + PM group maintained a uniform spherical structure, and numerous cells and ECM had been distributed among and within the microspheres (Fig. 6C). The short axis/long axis (S/L) value of microspheres represented the deformation degree of microspheres. The S/L value of microspheres in both groups showed no significant difference at 1 week, but it had become significantly higher in the mECM + PM group than the PM group at 4 weeks (Fig. 6D). Accordingly, the cell numbers in both the border and interior of the materials in the mECM and mECM + PM groups were significantly lower than those in the PM group after 1 week. At 4 weeks, the cell number in the interior of microspheres in the mECM + PM group was markedly higher than that in the PM group (Fig. 6E). Then, we evaluated the collagen deposition and distribution through Masson staining (Fig. 6F–H). At 1 week, loose collagen was observed at the boundary between tissue and materials in the mECM and mECM + PM groups, there was a small amount of collagen distribution around the microspheres in the PM group, with a thickness of 23.5 ± 5.4 μm (Fig. 6G). However, a large number of functional capillaries were detected inside collagen areas. The number of capillaries was significantly higher in the mECM group and mECM + PM group than that in the PM group (Fig. 6H). At 4 weeks, mECM hydrogel degraded and collagen was distributed among the microspheres. The thickness of collagen layers distributed at the boundary of the material and tissue

was 91.3 ± 23.9 μm in the PM group, and 213.4 ± 84.5 μm in the mECM + PM group, a large number of capillaries still existed inside the collagen in the PM and mECM + PM group, which was not a foreign body collagen encapsulation reaction (Fig. 6G, H). Moreover, the number of foreign body giant cells (FBGCs) was quantified with H&E staining images (Fig. 6I). At 1 week, the number of FBGCs was low, and showed no significant difference among all groups. At 4 weeks, the number of FBGCs slightly increased in both the PM group and the mECM + PM group, and it was significantly higher in the PM group than that in the mECM + PM group. No FBGCs were detectable in the mECM group because of the mECM degradation (Fig. 6I).

We next evaluated the immune response to the composites. Inducible nitric oxide synthase (iNOS)$^+$ proinflammatory macrophages were mainly distributed around the implanted materials, and only a small number of these cells had infiltrated into the interior of the mECM and microspheres after 1 week, and no significant differences were observed among the three groups. The number of iNOS$^+$ macrophages in the PM group and mECM + PM group declined to a remarkable extent by 4 weeks, exhibiting no significant differences between these two groups (Fig. 6J, L). Additionally, a large amount of CD206$^+$ macrophages were distributed around all three kinds of implanted materials. A small number of CD206$^+$ cells had infiltrated into the interior of materials at 1 week, and there was no significant difference among three groups. The number of CD206$^+$ cells in the PM group and mECM + PM group revealed a downward trend at 4 weeks, showing no significant difference between the two groups (Fig. 6K, M). We also performed immunofluorescence staining on leukocyte cells (CD45$^+$), T cells (CD3$^+$), and B cells (CD20$^+$), as exhibited in supplementary Fig.13. At 1 week, CD45$^+$ cells were mainly distributed at the boundary of materials in all groups, and it was significantly higher in the mECM group and mECM + PM group than that in the PM group. A small number of CD45$^+$ cells can infiltrate into the interior of materials, and the number of CD45$^+$ cells in the mECM group and mECM + PM group was higher than that in the PM group, showing no significant difference (Supplementary Figs. 13A, D). At 4 weeks, the number of CD45$^+$ cells significantly decreased, and there was no significant difference between the PM group and the mECM+PM group (Supplementary Fig. 13A,D). Furthermore, only a small number of CD3$^+$ T cells and CD20$^+$ B cells was distributed in the surrounding of the materials at 1 and 4 weeks, and there was no significant difference among these groups (Supplementary Fig. 13B, C, E, F). Additionally, the hemolytic rates of mECM + PM and mECM@IL4 + PM@IGF1 were less than 5%, which demonstrated that the materials exhibited high hemocompatibility (Supplementary Fig. 14).

**mECM@IL4 + PM@IGF1 composites enabled muscle regeneration after tibialis anterior volumetric muscle loss (VML) in rat models**
Rat models of volumetric muscle loss (VML) (40%) of the tibialis anterior were used to evaluate the pro-regenerative effects of different materials, including PM, mECM, mECM + PM, mECM@IL4 + PM, mECM + PM@IGF1 and mECM@IL4 + PM@IGF1 (Fig. 7A). The muscle

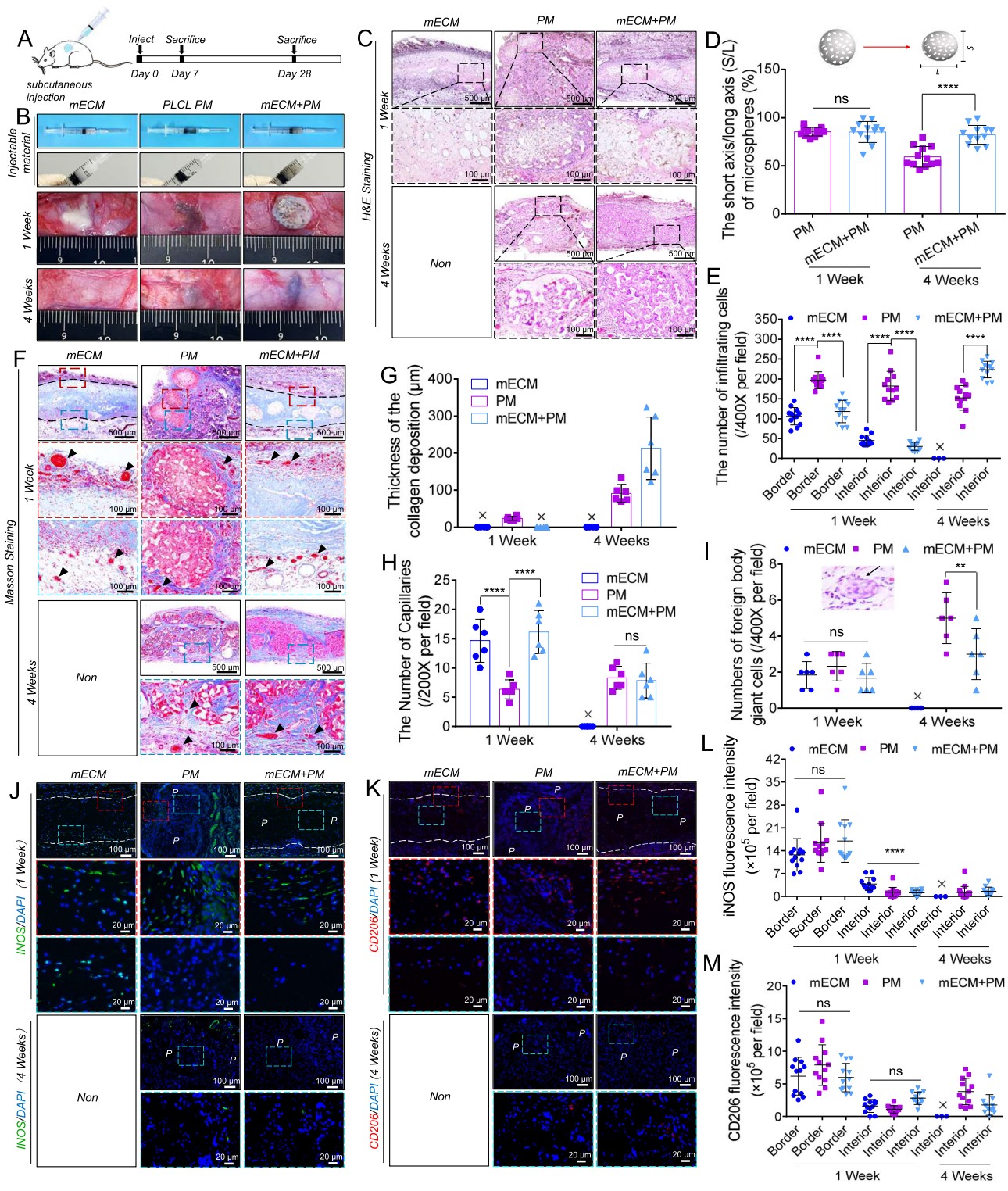

volume of the injured side was significantly smaller than that of the normal side, and injected microspheres could still be observed in the PM, mECM + PM, mECM@IL4 + PM, mECM + PM@IGF1 and mEC-M@IL4 + PM@IGF1 groups at 2 weeks. Moreover, the muscle volume of the injured defect side macroscopically increased in all groups at 8 weeks. There were a few visible microspheres in the PM, mECM + PM, mECM@IL4 + PM, mECM + PM@IGF1 and mECM@IL4 + PM@IGF1 groups at 8 weeks as regenerated muscle tissue occupied the original defect space (Fig. 7B). We next performed routine histological staining and immunostaining to evaluation the quality of muscle repair,

including H&E staining, Masson's trichrome staining, and desmin immunofluorescence staining (Fig. 7C-E). Then, we performed quantitative analyses of macroscopic and histological images (Fig. 7F–T). Muscle mass ratio of the injured side/normal site in the mEC-M@IL4 + PM@IGF1 group was higher than that in the untreated, PM, and mECM groups at 2 weeks, and showed a significant difference from the untreated group (Fig. 7F). Furthermore, muscle mass in all groups had obviously increased at 8 weeks, and there was a significant difference between the mECM@IL4 + PM@IGF1 group and untreated and PM groups (Fig. 7F). At 2 weeks, H&E staining revealed that a large

**Fig. 6 | Injectability and biocompatibility of mECM + PM composites.**
**A** Schematic diagram of subcutaneous injection of composites in rat models and explanations at different time points. **B** The distribution of mECM, PM, and mECM + PM in syringes, and material retention after injection for 1 and 4 weeks. **C** H&E staining showing tissue ingrowth. Black dotted box: high magnification images. **D** S/L values of microspheres in the PM and mECM + PM groups at 1 and 4 weeks after implantation. Data are presented as mean ± SD ($n = 3$ biologically independent samples, 4 random fields per sample). ****$P < 0.0001$, ns: no significant difference, one-way ANOVA, multiple comparisons. **E** Statistical results of cellular infiltration at 1 and 4 weeks. Data are presented as mean ± SD ($n = 3$ biologically independent samples, 4 random fields per sample). ****$P < 0.0001$, one-way ANOVA, multiple comparisons. **F** Masson staining showing the collagen deposition. Black arrows: capillaries; white dashed line: the boundary between material and tissues; red and blue dotted boxes: high magnification images. **G** The thickness of the collagen layer. Data are presented as mean ± SD ($n = 3$ biologically independent

samples, 2 random fields per sample). **H** The number of capillaries in (**F**). Data are presented as mean ± SD ($n = 3$ biologically independent samples, 2 random fields per sample). ****$P < 0.0001$, ns: no significant difference, one-way ANOVA, multiple comparisons. **I** Statistical results of foreign body giant cells (FBGCs) in different groups at 1 and 4 weeks in (**C**). Data are presented as mean ± SD ($n = 3$ biologically independent samples, 2 random fields per sample). **$P < 0.01$, ns: no significant difference, one-way ANOVA, multiple comparisons. **J**, **K** Immunofluorescence staining images of iNOS (green) and CD206 (red) in different groups 1 and 4 weeks after subcutaneous injection. Nuclei: blue; P: porous microspheres; white dashed line: the boundary between material and tissues; blue dotted box: central region; red dotted box: marginal region. **L**, **M** Statistical analysis of the fluorescence intensity of iNOS and CD206. Data are presented as mean ± SD ($n = 3$ biologically independent samples, 4 random fields per sample). ****$P < 0.0001$, ns: no significant difference, one-way ANOVA, multiple comparisons.

---

amount of inflammatory and non-myogenic cells had infiltrated the defect in all groups. Porous microspheres were filled with host cells in the PM, mECM + PM, mECM@IL4 + PM, mECM + PM@IGF1 and mECM@IL4 + PM@IGF1 groups (Fig. 7C). At 8 weeks, a large degree of VML could still be observed in the untreated group, whereas myofibers had filled the defect site in the PM and mECM groups and the density of myofibers in the mECM group was higher than that in the PM group (Fig. 7C). More importantly, a high density of myofibers and partially degraded microspheres filled with neomuscle tissues were distributed within the defect site in mECM + PM, mECM@IL4 + PM, mECM + PM@IGF1, and mECM@IL4 + PM@IGF1 groups (Fig. 7C). Masson's trichrome staining showed that a large number of collagen fibers were deposited in the defect site in all groups at 2 weeks. Collagen deposition had decreased and was accompanied by an increased number of newly formed myofibers at 8 weeks (Fig. 7D). The statistical results also confirmed that the collagen density in each group at 8 weeks was lower compared with 2 weeks. Moreover, the collagen density in the mECM@IL4 + PM@IGF1 group was significantly lower than that in the untreated, PM, mECM, and mECM + PM groups at 8 weeks. (Fig. 7D, G). Desmin staining showed that a small number of neo-myofibers were distributed around the materials at 2 weeks, and only a few desmin+ cells were observed in the interior of microspheres in the PM, mECM + PM, mECM@IL4 + PM, mECM + PM@IGF1, and mECM@IL4 + PM@IGF1 groups (Fig. 7E). The statistics also confirmed that the desmin fluorescence intensity in the mECM@IL4 + PM@IGF1 group was higher than that in the untreated, PM, mECM, mECM + PM, mECM@IL4 + PM, and mECM + PM@IGF1 groups, and only displayed significant differences compared with the untreated and PM groups at 2 weeks (Fig. 7H). Moreover, we investigated the distribution of the cross-sectional area (CSA) of myofibers. The CSA of neomyofibers in the injured area was distributed in a small region in all groups at 2 weeks. Although the CSA curve in the mECM@IL4 + PM@IGF1 group moved towards the right compared with the other groups, it was still far from that of the normal group (Fig. 7I–N). At 8 weeks, the density of myofibers with large diameters greatly increased compared with 2 weeks in all groups. The mECM@IL4 + PM@IGF1 group produced the largest density of thick myofibers, but the quantity was still lower than that in the normal group. Notably, the number of desmin+ cells distributed in the interior of microspheres in the mECM@IL4 + PM@IGF1 group was higher than that in the PM, mECM + PM and mECM@IL4 + PM groups (Fig. 7E). Furthermore, the expression of desmin in the mECM@IL4 + PM@IGF1 group was close to normal muscle tissue and higher than that in the other groups, showing significant differences compared to the untreated, PM, mECM + PM and mECM@IL4 + PM groups (Fig. 7I). At 8 weeks, the CSA distribution curve in all groups shifted to the right, and the distribution pattern of the mECM@IL4 + PM@IGF1 group was similar to the normal distribution and the CSA of myofibers was also closer to that in the normal group (Fig. 7Q–T).

## mECM@IL4 + PM@IGF1 composites promoted vascularization and neuralization of regenerated muscle

Vascularization and neuralization are critical indicators to evaluate the functional regeneration of neomuscle tissue[43]. We first monitored blood flow recovery at the implantation site by laser doppler at 8 weeks. The implanted materials improved the mean perfusion of the injured muscle site compared with the untreated group. The statistics displayed that the mean perfusion in the mECM@IL4 + PM@IGF1 group was significantly higher than that in the untreated, PM, mECM, mECM + PM, and mECM@IL4 + PM groups, although it was still lower than that in the normal group (Fig. 8A, B). H&E staining was used to evaluate functional capillary formation, displaying red blood cells in the capillary lumens (Fig. 8C). At 2 weeks, material implantation significantly increased the infiltration of blood capillaries into the injured area, and most of them distributed among the microspheres and inside the hydrogels, whereas only a few had distributed within the microsphere interiors. According to the statistical analysis, the number of functional capillaries in the mECM@IL4 + PM@IGF1 group was significantly higher than that in the untreated, PM, mECM, mECM + PM and mECM@IL4 + PM groups (Fig. 8C, D). Notably, blood capillaries were distributed not only in the neomuscle regeneration areas but also inside the microspheres at 8 weeks. The number of capillaries in composites groups (mECM + PM, mECM@IL4 + PM, mECM + PM@IGF1 and mECM@IL4 + PM@IGF1) were significantly lower than that in the PM, mECM groups, and approached to normal group (Fig. 8C, D). The presence of α-SMA+ pericytes is a marker of mature blood capillaries[44]. At 2 weeks, the number of α-SMA+ mature blood capillaries in the mECM@IL4 + PM@IGF1 group was significantly higher than that in the untreated, PM, mECM, mECM + PM and mECM@IL4 + PM groups. However, these numbers decreased in all groups at 8 weeks, and showed no statistical differences among all groups (Fig. 8E, F).

We next performed electrophysiological testing to determine the neural response functions of the neomuscle after different treatments at 8 weeks. The maximum amplitude of the CAMP in the mECM@IL4 + PM@IGF1 group was higher than other groups, and displayed a significant difference from that of untreated, PM and mECM groups, although it was still lower than that in the normal group (Fig. 8G, H). As a nerve fiber marker, NF-09 immunofluorescence staining showed that only a small number of NF-09+ nerve fibers scattered in the marginal area of the muscle defect, and there was no NF-09 positive expression in the interior of microspheres at 2 weeks. Statistically, NF-09+ expression in the mECM@IL4 + PM@IGF1 group was higher than that in the untreated, PM, mECM, mECM + PM, mECM@IL4 + PM and mECM + PM@IGF1 groups, and displayed significant differences compared with the untreated group. At 8 weeks, NF-09+ expression increased in all groups, and NF-09+ in the mECM@IL4 + PM@IGF1 group showed nerve fiber bundling patterns similar to that in natural muscle

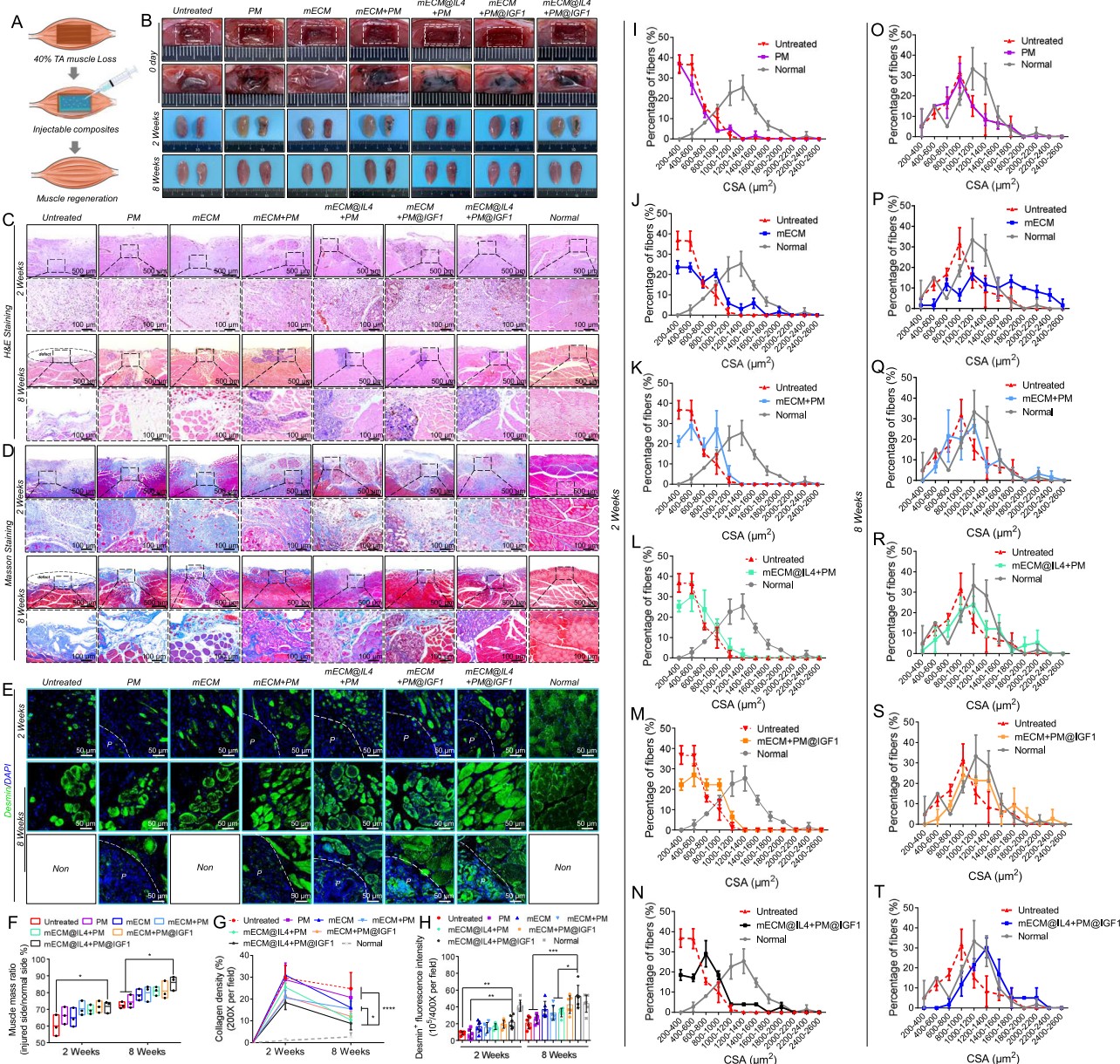

**Fig. 7 | mECM@IL4 + PM@IGF1 composites promote muscle regeneration in the volumetric muscle loss (VML) model of rats at 2 and 8 weeks. A** Schematic diagram of muscle regeneration enhanced by the injectable composites. **B** Macroscopic images of VML treated with different materials on day 0, and muscle regeneration at 2 and 8 weeks post-treatment. **C–E** H&E, Masson staining and desmin (green) immunofluorescence images showing muscle regeneration. Nuclei in (**E**): blue; black dotted box: high magnification images; P: porous microspheres; white dashed line: the boundary between microspheres and tissues. **F** The muscle mass ratio of the injured side/normal side. The floating bars represent the range of minimum to maximum values, and the line inside the box represents the mean value ($n = 3$ biologically independent samples). *$P < 0.05$, one-way ANOVA, multiple comparisons. **G** Statistical analysis of collagen deposition in (**D**). Data are presented as mean ± SD ($n = 3$ biologically independent samples, three random fields per sample). ****$P < 0.0001$, *$P < 0.05$, one-way ANOVA, multiple comparisons. **H** Statistical analysis of desmin fluorescence intensity in (**E**). Data are presented as mean ± SD ($n = 3$ biologically independent samples, 2 random fields per sample). ***$P < 0.001$, **$P < 0.01$, *$P < 0.05$, one-way ANOVA, multiple comparisons. **I–N, O–T** Frequency distribution statistics of the cross-sectional area (CSA) of neonatal myofibers at 2 and 8 weeks. Data are presented as mean ± SD ($n = 3$ biologically independent samples, 25 random myofiber areas per sample).

(Fig. 8I), and it was significantly higher than that in the untreated, PM, mECM, mECM + PM and mECM@IL4 + PM groups (Fig. 8J).

## mECM@IL4 + PM@IGF1 composites switched macrophage polarization in vivo

Finally, we evaluated the effect of composites on the number and distribution of pro-inflammatory and pro-regenerative macrophages (Fig. 9). iNOS, as a proinflammatory macrophage marker, had little expression in the normal group at 2 weeks, whereas it was highly expressed in the other groups. iNOS+ cells were distributed both inside

mECM hydrogels and microspheres. The expression of iNOS in the mECM@IL4 + PM@IGF1 group was lower than that in the other groups, and displayed a significant difference from that in the untreated group (Fig. 9A, B). At 8 weeks, the expression of iNOS decreased in all groups, and the number of iNOS+ cells in the mECM@IL4 + PM@IGF1 group was significantly lower than that in the untreated, mECM + PM and mECM@IL4 + PM groups (Fig. 9A, B). TNF-α, a pro-inflammatory cell secretory factor, was expressed inside the mECM hydrogel and microsphere at 2 weeks. TNF-α+ expression in the mECM@IL4 + PM@IGF1 group was lower than in the other groups, showing a

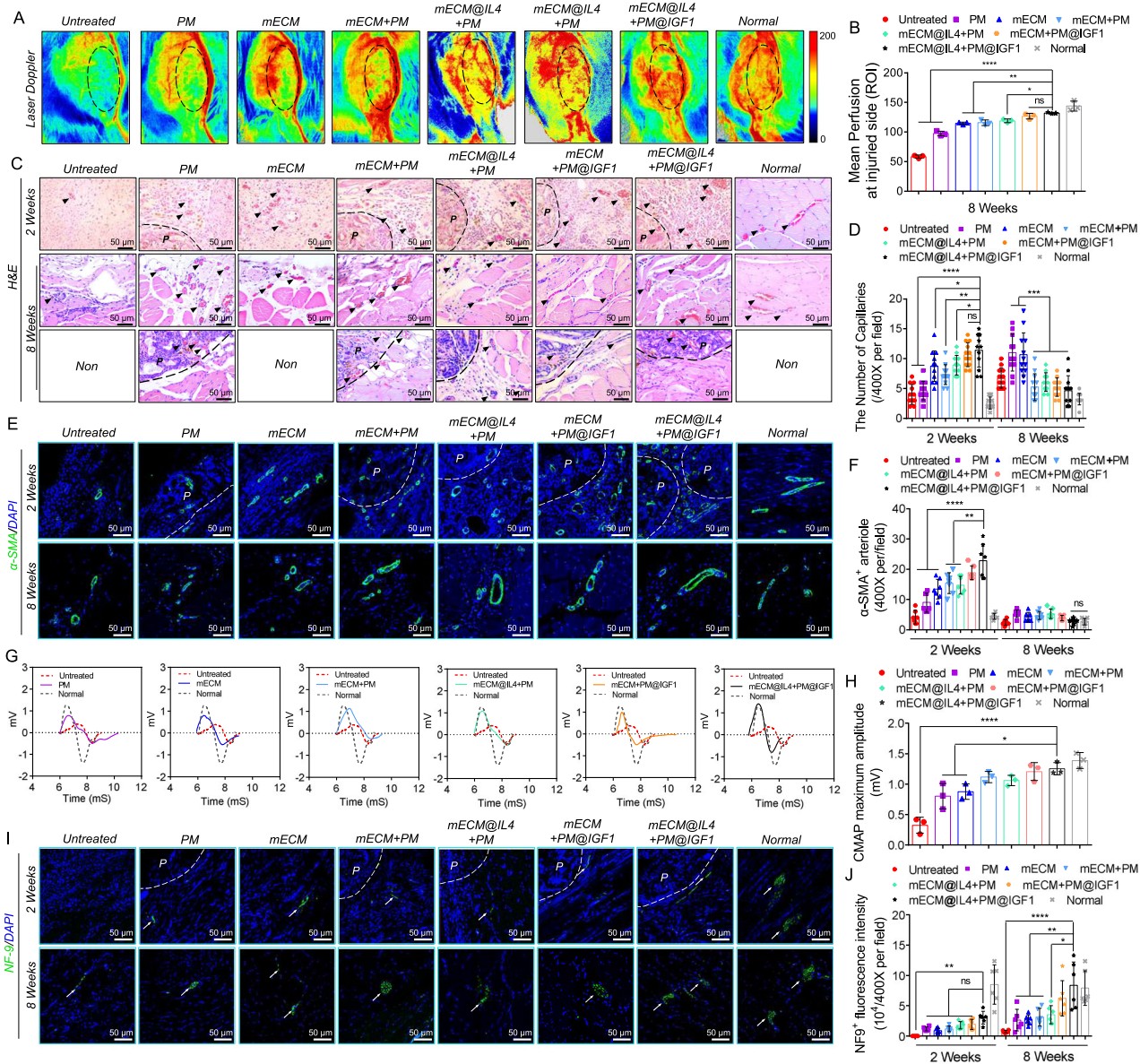

**Fig. 8 | mECM@IL4 + PM@IGF1 composites promote vascularization and neuralization of neomuscle in the TA defect model of rats at 2 and 8 weeks.** **A** Laser doppler of blood flow recovery after different treatments for 8 weeks. Black dashed line: muscle defect areas. **B** Statistical results of perfusion. Data are presented as mean ± SD ($n = 3$ biologically independent samples). ****$P < 0.0001$, **$P < 0.01$, *$P < 0.05$, ns: no significant difference, one-way ANOVA, multiple comparisons. **C** H&E staining showing functional capillary. Black dashed line: the boundary between microspheres and tissues; black triangular arrows: capillary. **D** The number of capillaries in (**C**). Data are presented as mean ± SD ($n = 3$ biologically independent samples, 4 random fields per sample). ****$P < 0.0001$, ***$P < 0.001$, **$P < 0.01$, *$P < 0.05$, ns: no significant difference, one-way ANOVA, multiple comparisons. **E** α-SMA (green) immunofluorescence images showing mature capillary. Nuclei: blue; P: porous microspheres; white dashed line: the

boundary between microspheres and tissues. **F** Statistical results of mature capillaries. Data are presented as mean ± SD ($n = 3$ biologically independent samples, 2 random fields per sample). ****$P < 0.0001$, **$P < 0.01$, ns: no significant difference, one-way ANOVA, multiple comparisons. **G** Representative patterns of CMAP after different treatments at 8 weeks. **H** Quantitative analysis of CMAP maximum amplitudes. Data are presented as mean ± SD ($n = 3$ biologically independent samples). ****$P < 0.0001$, *$P < 0.05$, one-way ANOVA, multiple comparisons. **I** NF-09 (green) staining images showing neuralization (white). Nuclei: blue; P: porous microspheres; white dashed line: the boundary between microspheres and tissues. **J** The statistical results of NF-09+ fluorescence intensity. Data are presented as mean ± SD ($n = 3$ biologically independent samples, 2 random fields per sample). ****$P < 0.0001$, **$P < 0.01$, *$P < 0.05$, ns: no significant difference, one-way ANOVA, multiple comparisons.

significant difference from the untreated and PM groups. At 8 weeks, TNF-α+ decreased obviously in all groups, with only a small amount of expression in the untreated group and almost no expression in the other groups (Fig. 9C, D). CD206, a pro-regenerative macrophage marker, distributed both inside microspheres and hydrogels. CD206+ expression in the mECM@IL4 + PM@IGF1 group was significantly higher than that in the other groups at 2 weeks. Notably, CD206+ expression was almost absent in the untreated and mECM groups at 8 weeks, while still exhibiting high expression inside microspheres in

the PM, mECM + PM, mECM@IL4 + PM, mECM + PM@IGF1 and mEC-M@IL4 + PM@IGF1 groups. Additionally, the expression of CD206 in the mECM@IL4 + PM@IGF1 group was significantly higher than that in the PM, mECM + PM, and mECM@IL4 + PM groups (Fig. 9E, F).

## Discussion
In this study, we constructed an injectable composite composed of elastic porous PLCL microspheres and mECM hydrogels, which were loaded with IGF-1 an IL-4, respectively. We demonstrated the

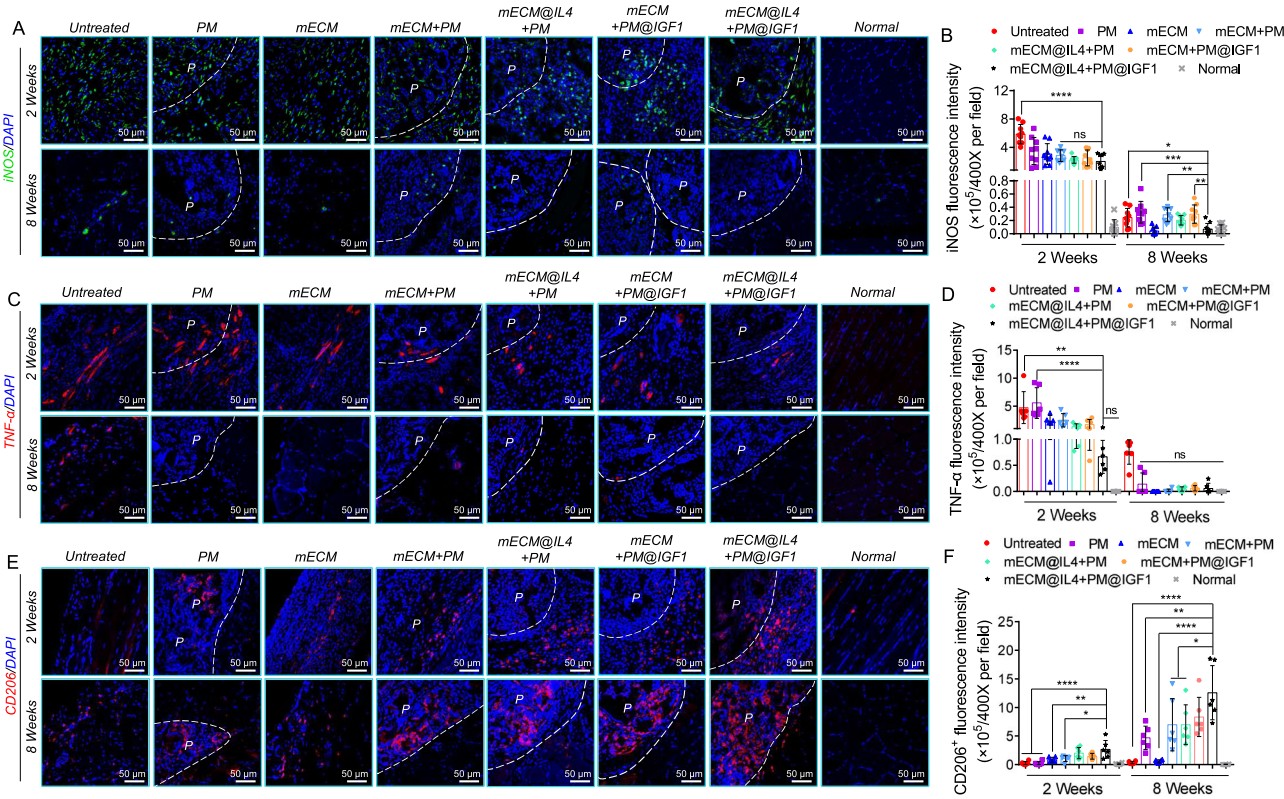

**Fig. 9 | mECM@IL4 + PM@IGF1 composites regulated the immune microenvironment in vivo at 2 and 8 weeks. A, C** Immunofluorescence images of the pro-inflammatory macrophage markers iNOS (green) and TNF-α (red). Nuclei: blue; P: porous microspheres region; white dashed line: the boundary between microspheres and tissues. **B, D** Fluorescence intensity statistics of iNOS and TNF-α. Data are presented as mean ± SD ($n = 3$ biologically independent samples, 3 random fields per sample). ****$P < 0.0001$, ***$P < 0.001$, **$P < 0.01$, *$P < 0.05$, ns: no significant difference, one-way ANOVA, multiple comparisons. **E** Immunofluorescence images of the pro-regenerative macrophage marker CD206 (red). Nuclei: blue; P: porous microspheres region; white dashed line: the boundary between microspheres and tissues. **F** Statistical plots of CD206 fluorescence intensity. Data are presented as mean ± SD ($n = 3$ biologically independent samples, 2 random fields per sample). ****$P < 0.0001$, **$P < 0.01$, *$P < 0.05$, one-way ANOVA, multiple comparisons.

injectability and biocompatibility of the composites, as well as the capability of promotively regulating cell behavior and promoting endogenous tissue regeneration. The development and application of innovative injectable functional biomaterials can improve the therapeutic effect of minimally invasive surgery. Numerous studies have shown that ECM hydrogels possess high biocompatibility but lack mechanical strength and degrade rapidly[6,23,45]. Synthetic polymer porous microspheres can guide tissue regeneration[10,19,46], but their degradation byproducts easily trigger a local inflammatory response, limiting tissue regeneration[21]. Here, we overcame these shortcomings by combining the advantageous mechanophysical and cell supporting properties of porous microspheres with the injectability and biocompatibility of mECM hydrogels. During tissue repair, immune cells, tissue-specific cells, and stem/progenitor cells successively infiltrate the injury site and participate in tissue remodeling[1]. Therefore, modulating these cellular behaviors using biomaterial-loaded active factors is expected to synergistically promote tissue repair. Thus, we realized the dual-release of IL-4 and IGF-1 to co-regulate macrophages and tissue-specific cells, respectively, and to achieve timely tissue regeneration.

Synthetic polymeric porous microspheres is usually applied as injectable material to repair irregular tissue defects. Synthetic polymers, such as PLGA, PCL, PLA, etc., have been frequently employed to prepare porous microspheres[5,21], but the resultant microspheres are prone to collapse or deformation after injection. These deficiencies are rarely reported. However, PLCL materials have inherent flexibility, elasticity and tensile properties that out-perform PLGA and PCL, lending to the use of PLCL as an appropriate polymer choice for scaffold implantation in areas of tissue compression, tension, and flexibility, such as nerve conduits[47], artificial blood vessels[41], and skin grafts[48]. Some studies have constructed composite systems[49,50], but these composites were mainly designed for drug delivery, and did not take into account the effects of pore structure, elasticity, material compatibility, and complementarity of synthetic polymer and ECM on in vitro injection and in vivo tissue regeneration. In this study, we selected PLCL to construct elastic PM to allow for the recovery of original shape after pressing or injection. PLGA and PCL microspheres collapsed and broke or deformed after the compression tests, and thus were ruled out as candidate microsphere materials (Fig. 2G, I, Supplementary Fig. 3). In addition, the interconnected pore structure of microspheres has been proven to significantly promote cell infiltration and tissue ingrowth compared with low porosity or non-PM[46]. However, the mechanical strength of porous microspheres is typically poor, due to the intrinsic properties of synthetic polymers and the necessity for large pore structures[46,51]. Few studies have tested the spatial structural changes of porous microspheres after injection or under shear stress. Thus, we gave special focus to improve the mechanical properties of polymer-based porous microspheres. The result of implementing a combination of microsphere synthesis technologies cumulated in elastic PLCL porous microspheres that was capable of maintaining highly interconnected pore structures and drug delivery efficiency even after injection or compression. This resulted in maximizing the availability of niches for cell recruitment, attachment, and differentiation, as well as drug delivery system, thereby improving outcomes of guiding tissue regeneration. Furthermore, the addition of elastic microspheres in the composites

substantially enhanced the mechanical strength of the injectable mECM hydrogel (Fig. 3I–L, Supplementary Figs. 5B–E, 7B–E). Moreover, the combinatory strategy we developed offers an alternative to pre-prepared scaffolds or materials with similar functions, but provides an injectable platform that can meet the needs of difficult to reach areas that are inaccessible to prepared scaffolds whilst requiring appreciably less time spent in surgical operations.

Notably, during subcutaneous injection in rats, the microspheres alone were prone to aggregation and sedimentation, and this made it difficult to achieve in situ retention and homogenous distribution after injection. In the composite system, as the microspheres were uniformly distributed inside the mECM hydrogel, aggregation or loss was effectively avoided during the injection process, and in situ gelation achieved after injection set the interconnected microsphere-ECM network in place (Supplementary movies 8-10). At 4 weeks after subcutaneous injection, the mECM hydrogel was degraded entirely, while the PM in the composites were able to maintain their three-dimensional shape and microscopic pore structure, which had been infiltrated by a large number of non-inflammatory cells and ECM. These indicated that PM in the composite system could compensate for the disadvantage of rapid degradation of mECM hydrogels, and continuously recruited and supported endogenous cells to promote tissue regeneration at the defect site (Fig. 6C, E). Additionally, the composites did not cause foreign body reaction. Although collagen deposition was observed around the materials, a large number of functional capillaries distributed inside the collagen deposition areas, which can promote substance exchange between surrounding tissues and infiltrated tissue inside the materials, as well as integration of implanted material with surrounding tissues (Fig. 6F–H). The composites possessed high biocompatibility and realized the complementation of advantages that can come from the application of different materials for in situ tissue regeneration after the successful implementation of minimally invasive procedures.

Autologous local muscle flap transplantation is the "gold standard" for the clinical treatment of VML. However, the suitable autografts are limited, the flap transplantation can lead to complications such as donor site morbidity and infection[52]. Up to now, there is no clinically available products for VML treatment. In a clinical study, Badylak, etc. used xenogeneic acellular materials (MtriStem, BioDesign, and XenMatrix) to repair VML, which could increase muscle mass and functional recovery[53]. These types of products are mainly ECM scaffolds derived from porcine dermis, bladder, or small intestinal submucosal, which lack tissue specificity. These materials have no injectability. Furthermore, muscle regeneration is a biological process mediated by various kinds of cells in sequence, and the coordinated modulation of macrophage phenotype and stem/progenitor cell behavior can promote VML repair[1]. We verified the capability of composite materials loaded with IL-4 and IGF-1 to promote endogenous tissue regeneration in a rat VWL model, the pro-regenerative effect of the dual-factor composites was stronger than that of the non-factor and the single-factor composites. The mECM@IL4 + PM@IGF1 composites were able to modulate macrophage phenotypic polarization in vivo, which in turn enhanced myofiber formation, vascularization and innervation (Figs. 7–9). We speculate that the composites promoted myofiber regeneration mainly through the following two aspects. First, the effect of IL4 and IGF-1 in mECM@IL4 + PM@IGF1 composites synergistically increased the proportion of pro-regenerative macrophage phenotypes, which not only increased the expression of the anti-inflammatory factors IL-4 and IL-10, but also decreased the expression of the anti-inflammatory factors IL-1β, IL-6, IL-1α, IL-12P70, MCP-1, IFN-γ, TNF-α and IL-5 (Fig. 4I-N, Supplementary Fig. 10). This paracrine effect of macrophages promoted CTX-injured myosatellite cell differentiation (Fig. 5G–K). Work from the lab of Prof. Nenad Bursac[54] demonstrated that the paracrine effects of macrophages protected myogenic cells from further damage by activating

anti-apoptotic pathways and inhibiting caspase-3 activation. Furthermore, ECM itself has been proven to promote muscle regeneration by regulating immune cell behaviors. Kaitlyn et al. showed that ECM activated mTOR/Rictor-dependent Th2 signaling pathway to induce IL-4-dependent macrophage polarization and promoted damaged muscle regeneration by regulating the adaptive immune system[55]. In addition, ECM was also able to stimulate endogenous NK cells to produce chemokine Xcl-1 and recruit DC cells to promote the secretion of cytokines such as IL-4, IL-10, and IL-33 in defect sites, thereby improving the myofibers regeneration[56]. However, the capability of ECM scaffolds to recruit and support resident myogenic stem cell ingrowth and to promote tissue regeneration is still limited. The second aspect was the controlled release of IGF-1 sustained the activation of myosatellite cell proliferation and differentiation, myofiber survival (Fig. 5, Supplementary Fig. 12), and hypertrophy-related signaling pathways[38,57–60], such as the PI3K/Akt[57,59–61], IGF-1[58] and mTOR[61] pathways, thus promoting myofiber formation. In addition, numerous studies have shown that pro-repair macrophage paracrine factors can also encourage endothelialization[62], vascularization[63], and neural regeneration[64], which are consistent features also demonstrated in our study.

The injectability and in situ pro-regenerative potential of our functionalized composite system was validated in a small animal model. However, further validation is still needed in larger models and under pathogenic conditions of disease models. The effect of elastic porous microspheres combined with other select tissue-specific ECM for the repair of distinct tissue types after injury or surgical traumas, especially tissues under mechanical load such as articular cartilage and intervertebral discs, is a natural future direction to explore the scope of our developed composite system. Cell-laden porous microspheres strategies were used to enhance tissue regeneration[11,65], but it was noted that shear stress during injection had consequential outcomes on microsphere integrity and unavoidable cell apoptosis or loss. Therefore, pre-seeding of elastic porous PLCL microspheres with tissue-specific or stem/progenitor cells in combination with the support of an ECM hydrogel carrier may offer an alternative method for improving cell survivability post-implantation. Also, the introduction of hydroxyl, amino, polypeptide (RGD) on the surface of the elastic microspheres is expected to improve the efficiency of cell adhesion and drug delivery, which are worthy of further study. In this work, we constructed functionalized injectable composites, which employed the complementary advantages of two different material structures and the synergistic effect of dual bio-factors to achieve endogenous tissue regeneration. We demonstrated the injectability, biocompatibility, and in situ pro-regenerative potential of the composites in vitro and in vivo. Furthermore, the regenerative potential of composites for other tissue defects, including nucleus pulposus, cartilage etc., through minimally invasive therapeutic manner is worthy of investigation in the future. Our study also opens a new avenue for designing and fabricating novel injectable bioactive materials.

## Methods

### Ethical regulations statement
All animal experiments were approved by the animal experiments ethical committee of Nankai University, Tianjin, China (2022-SYDWLL-000432) and met the requirements of the Guidelines for Care and Use of Laboratory Animals.

### Animal experiments
Seventy-five male Sprague Dawley rats (aged 8–10 weeks with the weight range of 280–300 g), 30 male Sprague Dawley rats (aged 4–5 weeks with the weight range of 120–150 g) and 42 male Sprague Dawley mammary rats (5 days) were used in this study. All animals were purchased from SPF Biotechnology Co., Ltd. (Beijing, China). Rats were randomly divided into cages (3 rats per cage) and adapted to a

pellet-based diet with tap water for 7 days in an SPF environment. The ambient temperature was maintained at $23 \pm 3\,°C$, humidity at 40–70%, and cycle light/dark for 12 h. Rats were fed a standard food diet and tap water. And during the feeding period, the animal room was cleaned regularly. Rats were fasting 12 h pre-surgery but can have water. After surgery, all experimental animals were lying sideways on a heating blanket ($37\,°C$) to maintain their smooth breathing and body temperature within 2 h. Paid attention to maintaining a quiet environment until the animals were completely awake. All experimental animals were fasting 12 h post-surgery but can drink freely. During the wound recovery period, we also paid attention to animals' physical conditions, such as feeding, excretion, surgical infection, and suture status. In addition, we recorded the local reaction of the injection site after injecting different materials in the rat subcutaneous injection model and VML model, including visible swelling, redness, edema, and abnormal color. Sixty-six rats were used to construct VML models. Nine rats were used to evaluate the cell infiltration and immunomodulatory properties of composites. 42 male Sprague Dawley rats (120–150 g) were used to isolate primary macrophages. All mammary rats were used to isolate primary muscle satellite cells. For all experiments $n = 3$ rats per group was set for obtaining numerical data.

## Materials

Poly (l-lactide-co-ε-caprolactone) (PLCL) (50:50) with viscosity of 2.6–2.8 and poly (lactic acid/glycolic acid) (PLGA) (50:50) with molecular weight of 50 kDa were purchased from Daigang Biomaterial Co., Ltd (Jinan, China), poly(caprolactone) (PCL) with molecular weight of 80 kDa and gelatin (G9391) were purchased from Sigma (USA). Span-80 was obtained from JSENB International Trade Co., Ltd (Hong Kong, China). Polyvinyl alcohol (PVA) was purchased from Dehang Wuzhou Technology Co., Ltd (Beijing, China). Dopamine hydrochloride (98%) was obtained from Innochem Technology Co., Ltd (Beijing, China). Cardiotoxin (CTX) was purchased from Guchen Biotechnology Co., Ltd (Shanghai, China). Rat-derived IGF-1 and Interleukin-4 (IL-4) were purchased from Biogot Technology, co, Ltd (Nanjing, China). 24-well cell climbing films, sodium dodecyl sulfate (SDS), ribonuclease (RNase), deoxyribonuclease I (DNase I), α-Galactosidase (α-GAL) kit, type I collagenase (C8140), sodium citrate antigen retrieval solution (50×) and 4% paraformaldehyde solution were obtained from Solarbio (Beijing, China). DNA quantitative kit was purchased from Yeasen Biotechnology Co., Ltd (Shanghai, China). Rat IL-4 and IGF-1 Elisa kits were purchased from Aibotech Biotechnology co, Ltd (Wuhan, China). DAPI fluoromount-G was purchased from Southern Biotech (USA). Alcohol, dichloromethane, and xylene were purchased from Tianjin Chemical Reagent Company (Tianjin, China). 3-0 and 9-0 nylon needle sutures were purchased from Ningbo Medical Needle Co., Ltd (Zhejiang, China). 3 M Tegaderm™ films were purchased from Polite Trading Co., Ltd (Tianjin, China). Paraffin wax was purchased from Leica Biosystems Richmond (USA). Flou-8 AM fluorescence probe (C0012) was obtained from Applygen Gene Technology Co., Ltd (Beijing, China). The information on primary and secondary antibodies is exhibited in Supplementary Table 1.

## Preparation of porous polymeric microspheres and dopamine modification

Different porous polymer microspheres were prepared using a microfluidic device. The water-in-oil (W-O) emulsion was prepared through emulsifying 2 mL aqueous gelatin (2.5–10.0%) and 100 μL span-80 in a 7.5 mL different polymer solution (2%) using a cell ultrasonic-homogenizer (JY92-IIN, China) at 30% ultrasonic power for 1 min. Subsequently, the prepared emulsion was introduced as a discontinuous phase into the fluidic device, and aqueous PVA solution (1%) served as the continuous phase. Then, the water-in-oil-in-water (W-O-W) droplets were formed at the tip of the needle, and the polymer-gelatin microspheres were obtained in ice water and followed

by stirring at 75 r/min for 12 h to evaporate excess dichloromethane solvent, and the gelatin component was removed in water at $40\,°C$. Porous polymer microspheres with different diameters and pore sizes were obtained by adjusting the needle size (18G, 20G, 22G, 24G, 26G) and gelatin concentration (2.5%, 5.0%, 7.5%, and 10.0%).

Porous PLCL microspheres were added in 0.5 mg/mL dopamine tris-buffer (pH = 8.5), 75 r/min shaker for 12 h. After modification, the tris-buffer removed, and the microspheres were cleaned with sterile water three times.

## Characterization of the porous polymeric microspheres

Stereoscopic microscopy and scanning electron microscopy (SEM) were used to observe the macro- and microscopic structures of PM. Atomic force microscopy (AFM) was used to observe the roughness of porous PLCL microspheres before and after dopamine modification, and XPS spectroscopy was used to observe the $N\,1s$ peak at 398.4 eV. The pore areas were calculated in SEM images of PM using image por-plus, and the relative pore sizes were converted by the following formula.

$$S = \pi \left( d/2 \right)^2 \tag{1}$$

The porosity of the PM was measured by liquid displacement method. PM was immersed in the dehydrated ethanol (initial volume $V_0$) for 5 min. $V_1$ was defined as the total volume of the system when the microspheres were immersed in the dehydrated alcohol. Then, microspheres were removed and the volume of the residual ethanol was recorded as $V_2$. The porosity of PM was defined as the following equations:

$$\text{Porosity} = \frac{V_0 - V_2}{V_1 - V_2} \times 100\% \tag{2}$$

Polymer microspheres were loaded in a 2.5 mL syringe with a 21G needle and injected rapidly through the glass capillary tube with an inner diameter of 0.8–1.0 mm. The morphology images of different microspheres were captured through an inverted microscope after injection. The elasticity of different polymeric microspheres (PCL, PLGA, and PLCL) was tested using tweezers by performing a single press for 30 s and 2 min, and multiple press for 15 times. The morphological photographs were recorded through a stereoscopic microscope, and the results were analyzed using Image-Pro Plus.

## mECM hydrogel preparation

Adult porcine skeletal muscle was purchased from Tianjin Er-shang Yingbin Meat Food Co. Porcine skeletal muscle was cut into small pieces and eluted with 1% SDS for 72 h, and then washed in sterile water for 48 h. DNase and RNase mixture were used to remove DNA and RNA at constant shaking speed of 150 r/min for 6 h at $37\,°C$. After washing for 48 h, the decellularized samples were freeze-dried and ground into powder using a cryogenic tissue grinder (Shanghai Jingxin Industrial Development Co., Ltd, China). The decellularization process can also refer to our previous study[29,66]. Then, mECM powder was dissolved in 0.01 M HCl and digested with pepsin at a ratio of 10:1 for 48 h. After digestion, the pH was adjusted to 7.4. The mECM pre-gels were mixed with 10 × PBS solution of 10% of its total volume and incubated at $37\,°C$ to form hydrogels.

## mECM characterization

Quantitative analysis of the remaining DNA content of samples was performed using a Quant-iT PicoGreen Assay (Invitrogen, Grand Island, NY) according to the manufacturer's instructions. Specifically, DNA was extracted from native muscle and decellularized muscle ECM prior to quantification. After diluting the DNA of samples and fluorescent dye, 125 μL of gradient DNA standard solution and the test solution

were added to the 96-well plate, then 125 μL of the diluted DNA fluorescent dye and reacted for 5 min. A multifunctional microplate reader (excitation wavelength: 480 nm, emission wavelength: 520 nm) was used to measure the fluorescence intensity of samples and quantified by a standard curve of DNA. Furthermore, the remaining α-galactosidase (α-Gal) samples were performed using α-GAL activity detection kit (Solarbio) according to the manufacturer's instructions. H&E and Masson's histological staining was performed to observe the morphology of myofibers and collagen composition. DAPI fluorescence staining was used to detect cells before and after decellularization.

### Preparation and characterization of the mECM + PM composites

mECM and porous PLCL microspheres were physically mixed at different mass ratios (30:1, 10:1, 5:1, 3:1) to obtain the mECM + PM composites. Then, different mECM+PM composites were loaded into 2.5 mL syringes, the sedimentation of microspheres was recorded at different time points (0, 5 min, 30 min, 6 h, 24 h and 48 h). SEM was used to detect the distribution of PM inside the mECM hydrogel in the lyophilized state. Under wet-state conditions, porous PLCL microspheres were labeled with rhodamine, mECM hydrogels were labeled with FITC, and three-dimensional spatial structures of composites were observed by a confocal laser microscope (Leica, SP8, Germany). Rheological measurements of the pure mECM hydrogel and mECM + PM composites (different mass ratios or different porosity of microspheres) were carried out using a rheometer at different temperatures from 10 to 40 °C. The storage modulus ($G'$) and loss modulus ($G''$) were measured using flat plates (10 mm), the shear strain was set to 1%, and the shear frequency was set to 12 rad/s.

### Fabrication of the mECM@IL4 + PM@IGF1 composites and release profile

1.5 mg dopamine-modified porous PLCL microspheres were placed in 500 μL IGF-1 solution (500 ng/mL) under alkaline conditions, incubated for 24 h on a shaker at 4 °C, and then freeze-dried to obtain PM@IGF-1 microspheres, 4 μL of 25 μg/ml IL-4 (total 100 ng) was directly added into the 500 μL mECM pre-gel. Finally, 500 μL pre-gel was mixed with 1.5 mg microspheres to obtain the mECM@IL4 + PM@IGF1 composites. For the IL-4 and IGF-1 release assay, 500 μL functionalized composites were placed in 1 mL PBS solution and incubated on a shaker at 37 °C. At different time points (3 h, 12 h, 1 day, 2 days, 3 days, 5 days, 7 days, 10 days, 14 days, 20 days), 500 μL supernatant was collected and added with 500 μL PBS. The collected samples were stored at −80 °C. Finally, the release of IL-4 and IGF-1 was measured using ELISA kits. In brief, 100 μL sample diluent was added to the antibody-coated ELISA plate and incubated for 2 h. After cleaning with wash buffer, 100 μL biotin conjugate antibody was added and incubated for 1 h. Subsequently, 100 μL streptavidin-HRP solution was added and incubated for 30 min. After cleaning the ELISA plate, 100 μL TMB substrate was added and incubated for 20 min. Finally, 50 μL stop solution was directly added to stop reaction, and the absorbance value was measured at 450 nm.

### IGF-1 release behavior after injection and deformation

3 mg PM@IGF-1 microspheres were resuspended in 1 mL PBS solution for injection test. Then, the supernatants in injection group and non-injection group were collected to detect the release of IGF-1 after 24 h.

3 mg PM@IGF-1 microspheres were resuspended in 1 mL PBS and placed on a 12-well plate for parallel plate compression test. The supernatants were collected firstly at different time points (0, 12 h, 24 h, 72 h, 120 h, 168 h and 240 h). Then, a 20 g weight placed on a glass plate was used to compress the microspheres for 30 s. After collecting 500 μL supernatant, the fresh PBS was supplemented to 1 mL, and

cultivated at 37 °C. Finally, ELISA kit was used to detect the IGF-1 release.

### Primary cell isolation and culture

Primary bone marrow-derived macrophages (BMDMs): male SD rats (120–150 g) were sacrificed, the femur and tibia were cut, and all cells were flushed out with basic medium. Then, cell suspension solution was passed through 70 μm and 40 μm microporous filter membranes. After centrifugation at 160 × g for 5 min, the supernatant was removed, and 10 mL erythrocyte lysate was added for 10 min and shake every 5 min, and then collected cells were seeded on 75 mL culture flask with 15% inactivated fetal bovine serum after centrifugation at 160 × g for 5 min. Subsequently, suspended cells were collected after 3–4 days of culture. The suspended cells were centrifuged, collected and inoculated in a 75 mL flask and cultured by adding 20 ng/mL GM-CSF macrophage-stimulating factor. The culture medium was changed every 3 days, and BMDMs were obtained after 7 days.

Primary muscle satellite cell extraction and culture: male SD mammary rats (5 days) were sacrificed, and all leg muscles were cut into small pieces, then treated with 0.2% type I collagenase for 40 min, and centrifuged at 500 × g for 1 min. The supernatant was removed, incubated with trypsin containing 0.2% EDTA at 37 °C for 40 min, and then terminated digestion by adding complete medium. The remaining impurities were removed by passing through 70 μm and 40 μm filters. The cells were then centrifuged at 160 × g for 5 min and resuspended with 3 mL medium. 2 mL of 80% percoll isolate, 8 mL of 20% percoll isolate, and 3 mL cell resuspension were slowly added into the centrifuge tube in sequence, and centrifuged at 3400 × g for 6 min. Cells from the middle layer of 20% and 80% of the isolate were collected for inoculation, the suspended cells were reinoculated using the differential adhesion method for 1 h and 2 h, and the suspended cells were taken and reinoculated. Cells at 70% confluence were passaged to prevent the differentiation of muscle satellite cells.

### The regulation of macrophage polarization in vitro

The composites were cocultured with BMDMs using Transwells, the extracted primary macrophages were inoculated at $5 \times 10^4$ cells/well in the lower chamber of 24-well plates. Different composites (200 μL) containing 0.6 mg microspheres with additional 200 μL 1640 basal medium were added to the upper chamber of the transwells, namely mECM + PM, mECM@IL4 + PM, mECM + PM@IGF1 and mECM@IL4 + PM@IGF1, and 100 ng IL-4 was mixed in each IL-4 loaded composites. 200 μL 1640 basal medium was added into the upper chamber as negative group, and 200 μL 1640 basal medium containing 100 ng IL-4 was set as positive group. After coculture for 1 and 3 days, CD68 and CD206 co-staining was performed, the images were captured under a fluorescence microscope (Zeiss Axio Imager Z1, Germany), and the statistical analysis was performed using Image-Pro Plus and GraphPad 6.0.

Flow cytometric analysis: The extracted BMDMs were inoculated in six-well plates at $1 \times 10^5$ cells/well, and different materials were added to the upper chamber of the Transwell. 2 mL 1640 basal medium mixed with different composites (500 μL) containing 1.5 mg microspheres were added to the upper chamber of the transwells, namely mECM + PM, mECM@IL4 + PM, mECM + PM@IGF1 and mECM@IL4 + PM@IGF1, 100 ng IL-4 was loaded in each IL-4 delivery composites. 2 mL 1640 basal medium was added into the upper chamber as negative group, and 2 mL 1640 basal medium containing 100 ng IL-4 was set as positive group. After 1 and 3 days, the primary macrophages in the lower chamber were collected using a cell scraper and centrifuged at 160 × g for 5 min. The collected cells were incubated with Alexa Fluor® 647 CD206 antibody (1:200, 141712, Biolegend) and FITC® 488 CD68 antibody (1:100, 137012, Biolegend) for 30 min. Afterwards, the cells were washed twice with staining buffer, followed by resuspension in 800 μL staining buffer. The stained cells were detected using a

FACSCalibur flow cytometer (BD Biosciences) and CellQuest software (Pharminogen). Data were analyzed using FlowJo software.

Cytokine measurement: BMDMs at a density of $2.5 \times 10^5$ cells/well were seeded on the lower chamber of six-well plate transwells. BMDMs were first educated and cultured with 1640 medium containing 5% heat-inactivated FBS. Other cultivation conditions were consistent with flow cytometry analysis. After incubation for 1 and 3 days, the cell supernatants were collected and centrifuged at $250 \times g$ for 10 min, and stored at −80 °C. The samples were detected using magnetic bead (Luminex) multiplexed cytokine assays by Youning Weisheng Technology Co., Ltd (Shanghai, China).

### The synergistic effects of IL-4 and IGF-1 on the proliferation of myogenic cells

L6 cells (Rat, ATCC, CRL-1458) were seeded on 96-well plates at a density of $2 \times 10^3$ cells/well. After culturing for 12 h, the supernatant was removed, and 100 μL DMEM complete medium containing single factors (IL-4 or IGF-1) or dual factors (IL-4 + IGF-1) with different concertation were added and cultured for 1 and 3 days. At different time points, CCK-8 dilution in DMEM (1:10) was added to 96-plate for 40 min. The optical density (OD) value at 450 nm was detected with a microplate reader (Bio-Rad, USA) to evaluate cell proliferation.

### Interaction experiments of composites, BMDMs, and L6/muscle satellite cells

Cell viability assay: Different composites including PM@IGF1, mECM + PM, mECM + PM@IGF1 and mECM@IL4 + PM@IGF1 were added to 3 mL of DMEM medium and soaked for 1 day, and upper conditioned medium were centrifuged and collected. L6 cells were seeded on 96-well plates at a density of $2 \times 10^3$ cells/well. After culturing for 12 h, the supernatant was removed, and 100 μl of different conditioned medium containing 10% FBS were added and cultured for 1, 3, 5, and 7 days. 100 μl DMEM complete medium was added as negative control, and 100 ng/mL IGF-1 DMEM complete medium was set as positive control. At different time points, CCK-8 dilution in DMEM (1:10) was added to 96-plate for 40 min. The optical density (OD) value at 450 nm was detected with a microplate reader (Bio-Rad, USA) to evaluate cell viability.

Live/dead staining: The composites were cocultured with L6 cells using Transwells. L6 cells were inoculated at $1.0 \times 10^4$ cells/well on cell climbing films in the lower chamber of 24 wells. Different materials were added in the upper chamber of the transwells, including 200 μL DMEM basal medium, 200 μL mECM + PM composites, 200 μL DMEM basal medium mixed with 0.6 mg PM@IGF1, 200 μL mECM + PM@IGF1 composites, 200 μL mECM@IL4 + PM@IGF1 composites, and 200 μL DMEM basal medium containing 100 ng/mL IGF-1 as positive control. Dead/live staining was used to observe L6 cell survival after culturing for 3, 5, and 7 days, the images were captured under a fluorescence microscope (Zeiss Axio Imager Z1, Germany), the number of dead cells was calculated using Image-Pro Plus.

Apoptosis assay: L6 cells were inoculated at $1.0 \times 10^4$ cells/well in the lower chamber of 24-well plates under 2% low-serum culture, other cultivation conditions were consistent with live/dead staining. After 7 days of coculture, the cells were washed with PBS and fixed with 4% paraformaldehyde solution. Then, L6 cell apoptosis was detected using a one-step TUNEL Cy5 apoptosis detection kit (k1135, APE×BIO).

Primary muscle satellite cells injury model: To determine the appropriate injury conditions, satellite cells were treated with different CTX concentrations (0, 0.3 μM, 0.6 μM and 1.0 μM) for $2.0 \pm 0.5$ h or $4.0 \pm 0.5$ h. The CTX solutions were removed after treatment, and cells were treated with fluo-8am Ca$^+$ fluorescence probe for 30 min according to the manufacturer's instructions. Then, the intracellular Ca$^{2+}$ fluorescence intensity was detected at 490 nm using a multifunctional enzyme marker. In addition, Ca$^{2+}$ fluorescence images were

captured using a fluorescence microscope (Zeiss Axio Imager Z1, Germany). 0.3 μM CTX concentration was selected for subsequent experiment. Finally, we used the 24-well plate transwell to evaluate the effect of BMDMs cooperated with composites on the differentiation behavior of injured muscle satellite cells. Satellite cells were inoculated in the lower chamber at $2.5 \times 10^4$ cells/well for 3 days in advance. After 0.3 μM CTX treatment, BMDMs ($2 \times 10^4$ cells/well) were directly added in the lower chamber of the transwells and co-cultured with satellite cells, and composites were directly placed in the upper chamber which were named as BMDMs, composites and BMDMs/composites, respectively. Cells with and without CTX treatment were set as control group. Subsequently, DMEM medium containing 10% FBS and 2% horse serum added in the lower chamber was applied for cell differentiation. After 7 days of coculture, phalloidin (1:80, Solarbio, ca1620), desmin (1:200, Santa, sc23879) and CD206 (1:300, Abcam, ab64693) fluorescence co-staining was performed to observe the ability of primary muscle satellite cell differentiation, the fluorescence images were captured using a total internal reflection fluorescent microscope TIRF & Thunder. The myotube length and area, the nuclei number per myotube and the number of CD206$^+$ cells were analyzed using Image-Pro Plus.

### Hemolysis detection ex vivo

Fresh rat blood was collected and anticoagulated using 1% heparin sodium. 0.2 mL blood was added to the centrifuge tube containing 10 mL normal saline and different samples including mECM+PM composites (200 μL) and mECM@IL4 + PM@IGF1 composites (200 μL, 500 μL). 10 mL normal saline were used as the negative control group, and 10 mL deionized water was used as the positive control group. After incubating for 30 min, 4 mL of suspension in each group was collected and centrifuged at $1040 \times g$ for 5 min. The OD values of the supernatants (100 μL) were then measured using a UV/VIS spectrophotometer (UNIC 2802S, China) at 545 nm. The hemolysis rate was calculated using the following formula:

$$\text{Hemolysis rate}(\%) = \frac{OD_{545(\text{sample})} - OD_{545(\text{negative})}}{OD_{545(\text{positive})} - OD_{545(\text{negative})}} \times 100\% \qquad (3)$$

### Histocompatibility evaluation in subcutaneous injection model

Nine male Sprague–Dawley rats (280–300 g) were used to evaluate the immune response and cell infiltration of different materials. 1 mL mECM hydrogel, PM (3 mg) in 1 mL saline and 1 mL mECM + PM composites were prepared under sterile conditions and loaded into a 2.5 mL syringe, respectively. Then, they were subcutaneously injected into rats for 1 and 4 weeks. H&E staining was used to assess cell infiltration. Masson staining was used to assess the collagen deposition and distribution. To evaluate the immune response, immunofluorescence staining was performed to identify the immune cells with the following antibodies: CD68 (1:250, Abcam, ab31630), iNOS (1:300, Abcam, ab15323), CD206 (1:300, Abcam, ab64693), CD45 (1:150, Abcam, ab10558), CD20 (1:100, Abcam, ab64088) and CD3 (1:100, Abcam, ab16669). Subsequently, the following secondary antibodies were reacted with the corresponding primary antibodies: goat anti-rabbit IgG Alexa 594 (1:500, Invitrogen, USA) and goat anti-mouse IgG1 Alexa 488 (1:500, Invitrogen, USA). DAPI staining was used to label the cell nucleus (Southern Biotech, England). Finally, images were observed using a confocal laser microscope (Leica, SP8, Germany). The S/L value of microspheres, cell infiltration, and fluorescence intensity of iNOS and CD206 were measured using Image-Pro Plus 6.0 software.

### Construction of the rat VWL model

To prepare the VML models, adult male Sprague–Dawley rats (aged 8–10 weeks with a weight range of 280–300 g were randomly assigned

to the untreated (saline), PM, mECM, mECM + PM, mECM + PM@IGF1 and mECM@IL4 + PM@IGF1 composites groups. After anesthetization with ketamine (40 mg/kg)-xylazine (5 mg/kg)-acepromazine (1 mg/kg), and isoflurane gas was used as assisted anesthesia throughout the procedure. The hairs of rats' right legs were removed, and the skin was cut to expose the tibialis anterior muscle. The muscle defect with 10 mm × 6 mm ×4 mm defect accounting for a volumetric loss of ~40% of the tibialis anterior was resected using the surgical blade. After implantation of different materials, the clinical 3 M breathable patch was fixed to the defect site using 9-0 sutures (Lingqiao, Ningbo, China). The skin was closed with 3−0 monofilament nylon sutures (Lingqiao, Ningbo, China). All surgical procedures were performed in SPF environment, and all rats were fasting 12 h before and after surgery, but can have water. Finally, rats were euthanized at 2 and 8 weeks, and the regenerated muscle was obtained for histological and immunofluorescence staining analysis.

### Electrophysiology and blood flow recovery assessment
To characterize the electrophysiologic properties 8 weeks post-surgery, the rats were anesthetized, and the experimental muscle side was exposed. Then, stimulating electrodes combined with acupuncture needles were pierced into the two ends of the regenerated muscle, the compound muscle action potentials (CMAPs) were recorded using a 4-channel physiologic signal recorder (RM-6240, Chengdu Instrument Factory, Chengdu, China). The CMAP results were compared with the tibialis anterior muscle of the contralateral normal side.

Laser Doppler imaging was conducted to quantitatively detect blood flow recovery of neo-muscle at 8 weeks using a PeriCam PSI Information System (Perimed AB, Sweden). The sampling frequency was 60 Hz, and the sampling time was 25 frames per second. The average values of blood flow were subsequently determined.

### Histological staining
Muscle histological analysis was performed at 2 and 8 weeks after surgery. Briefly, the regenerated muscles of the experimental side and normal side were picked after euthanasia. Then, the explants were fixed in 4% paraformaldehyde at 4 °C for one day, then embedded in paraffin, cut into 5 µm sections with a microtome (Leica, Germany) and placed in 65 °C ovens for 6 h.

Before histological staining, muscle tissue sections were placed in xylene for 40 min to remove paraffin, followed by hydration using gradients of alcohol and immersion in distilled water for 5 min. Then, H&E and Masson staining were performed according to the kit instructions. Images were captured using a Leica microscope (Leica DM3000, Germany). The integrated optical density (IOD) of collagen fibers per unit area (mm$^2$) was measured using Image-Pro Plus 6.0 software.

### Immunofluorescence staining
After dewaxing and gradient hydration of the tissue sections, antigen repair was performed using sodium citrate solution, followed by blocking the nonspecific binding sites with 5% goat serum for 40 min and then incubating with different antibodies overnight at 4 °C, including Desmin rabbit polyclonal antibody (1:50, Abcam, ab15200), α-SMA rabbit polyclonal antibody (1:300, Abcam, ab7817), NF-09 mouse monoclonal antibody (1:200, Abcam, ab7794), iNOS rabbit polyclonal antibody (1:200, Abcam, ab15323), TNF-α rabbit polyclonal antibody (1:100, Abcam, ab183218), and rabbit CD206 polyclonal antibody (1:500, Abcam, ab64693). After washing with PBS, the sections were incubated with the corresponding secondary antibodies for 60 min at 37 °C. After washing with PBS 3 times, the tissue sections were stained with DAPI. The fluorescent images of the sections were captured with a confocal laser microscope (Leica, SP8, Germany). The quantification of fluorescence intensity was assessed using Image-Pro Plus 6.0 software.

### Statistical analysis
Each experiment was repeated at least three times independently with similar results. All data are expressed as the mean ± standard deviation (SD). GraphPad Prism 6.0 software (GraphPad Software Inc., La Jolla, CA, USA) was used for statistical analysis. Image pro-plus 6.0 software was used for semi-quantitative analysis of fluorescence intensity. At least three samples per group were randomly selected and measured in each group. Briefly, the single-channel fluorescence image was converted into a gray image at first, and the gray value of each pixel represents the fluorescence intensity. The software will select a threshold uniformly for measurement, and the integrated optical density (IOD) for specific area was quantified. For the measurement of microspheres diameter, pore size, and cell number, length, area, etc., the details of quantification were presented in main text. Single comparisons were performed using an unpaired Student's $t$ test. Multiple comparisons were performed using Tukey's post hoc test and one-way analysis of variance (ANOVA). Multivariant comparisons were performed using two-way ANOVA with Tukey's post hoc test. For all tests, $*p < 0.05$, $**p < 0.01$, $***p < 0.001$, $****p < 0.0001$ indicate statistical significance, ns: no significance.

### Reporting summary
Further information on research design is available in the Nature Portfolio Reporting Summary linked to this article.

## Data availability
All data needed to support the conclusions in the study are available within the article and/or the supplementary files and movies. Data underlying Figs. 1–9 and Supplementary Figs. 1–13 are provided with this paper in the source data file. Any additional requests for information can be directed to, and will be fulfilled by the corresponding authors. Source data files are provided with this paper. Source data are provided with this paper.

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

## Acknowledgements

This work was financially supported by the National Natural Science Foundation of China (NSFC) projects (81921004 (D.K.), 82272156 (M.Z.), 82202775 (W.L.)), CAMS Innovation Fund for Medical Sciences (no. 2022-I2M-1-023 (M.Z.)), the China Postdoctoral Science Foundation (BSMS7100011 (W.L.)). Inner Mongolia Autonomous Region Science and Technology Project (2022YFSH0063 (J.Z.)) and Beijing-Tianjin-Hebei Basic Research Cooperation Special Project (22JCZXJC00100 (M.Z.)).

## Author contributions

M.Z. and D.K. conceived the research; Y.L., M.Z., D.K., and W.L. designed the experiments; Y.L, S.L. and Y.W., performed the experiments; Y.L, S.L., Y.W., H.L., F.N., and Y.S., designed, fabricated and characterized composites; Y.L, J.Z., S.L., Y.W., and G.S., performed microsurgery of animal experiments. Y.L., M.Z., A.C.M., D.K., W.L., and Y.Z., interpreted the data, analysed the data and wrote the manuscript. All authors discussed the data and direction of the project at regular intervals throughout the study.

## Competing interests
