## [Peer Review File · Nature Communications]

Reviewers' Comments:

Reviewer #1:

Remarks to the Author:

This is a very interesting and innovative study on a timely and important topic in biomedical materials. The authors had constructed polymer microspheres with elastic characteristics, which filled the bottleneck problem of insufficient mechanical properties of porous microspheres. The developed mECM@IL4+PM@IGF1 multifunctional composites exhibited favorable injectability and biocompatibility, and achieved the regeneration of volumetric tibialis anterior muscle loss. In general, I think this work is interesting to readers and well-performed. I have some comments that need to be addressed by the authors.

1. The author said that synthetic polymers lack bioactivity, so how about the cytocompatibility of porous PLCL microspheres in vitro?
2. In Fig.3, the author needs to recheck the order of the figure legend number.
3. I suggest that the functionalization of microspheres with intracellular adhesion molecules rather than PDA/IGF-1 could be another direction. The authors could expand the discussion by referring to existing literature/ideas.
4. More details on the DNA and ELISA assays should be included in the method section.
5. Minor corrections I spotted so may as well mention:
 - 1) In Fig.1, "injection" should be "injection", "bioactive" should be "bioactive".
 - 2) Line 287, "IL--containing" should be "IL-4 containing".
 - 3) In Fig.4A, "MC-CSF" should be "GM-CSF".
 - 4) Line 354, "2.0±0.5 h" should be "2.0±0.5 hrs"
 - 5) In Fig.5, "3 days" should be "3 Days".

Reviewer #2:

Remarks to the Author:

In this work Li et al. describe the formulation of a porcine muscle ECM-derived hydrogel with embedded porous PLCL microspheres. By functionalizing the microspheres with IGF-1, and incorporating IL-4 in the ECM hydrogel, Li et al. demonstrate the controlled and sequential release of IL-4 then IGF-1 from the hydrogel composite. The authors nicely characterize the mechanical properties of the composite hydrogel, in a range that is appropriate for skeletal muscle, and demonstrate its injectability. Further, injection of the composite hydrogel, into a model of volumetric muscle loss in rats, resulted in improved muscle fiber histology, muscle perfusion, and functional innervation. In vitro studies were used to show the effect of the IL4/IGF1 composite hydrogels on macrophage polarization, and the effect of IGF1 on muscle cell growth.

While this is an interesting and versatile biomaterial for cytokine delivery, some additional data and discussion could be added to the manuscript to strengthen the significance and novelty of the work. ECM-based hydrogels as well as the immune polarizing and pro-regenerative effects of IL4 and IGF1 are well known in the field. Additional data showing how the embedded microspheres alter the mechanical properties of the composite (does microsphere pore size and/or sphere diameter change the properties of the composite? Can the density of microspheres be used to tune the mechanical properties of the composite?) would add to the versatility of the biomaterial and significance of the work. While Li et al. show controlled cytokine release, the release profiles were not explored in a therapeutic context. (Is there a regenerative advantage of sequential cytokine release, does cytokine order matter? Is the biomaterial needed or are soluble cytokines sufficient? The in vivo data in figures 7-9 do not contain cytokine only groups.) Lastly, the authors refer to synergy between IL4 and IGF1, but the animals were not treated with the cytokines individually in the muscle studies (figures 7-9). So it is unclear if the effects of the two cytokines are both required for muscle regeneration, and if their combination is additive or synergistic. Biomaterials loaded with individual growth factors would add to the work. Control groups in figures 7-9,

containing biomaterial without cytokines, suggest that the material itself has some beneficial effect on muscle regeneration. Additional discussion regarding the possible mechanism of the material itself would also strengthen the manuscript.

Comments

The authors should take greater care when describing comparisons between groups. Throughout the manuscript (and especially in the paragraphs starting on lines 266, and 456), the authors describe comparisons that are 'apparent,' or 'noticeable;' sometimes these differences are statistically significant and other times they are not. It would help the reader to critically follow the data if differences that are significant are consistently described as 'significant,' and those that are not statistically significant are explicitly described as 'not statistically significant.'

Examples: The authors should clarify that the difference between the control and ECM+PM group described in line 274 is not statistically significant. In lines 280 – 282 the authors state that the difference between the pure IL-4 group and the mECM@IL4+PM was 'noticeable,' however the data in Fig. 4H suggests that this difference is not statistically significant.

In lines 492 – 495, it should be made clear that the differences between the hydrogel+IGF1+IL4 group vs. the ECM only and ECM+PM groups are not statistically significant. (Alternatively, only reference the significant differences in the text).

In line 562 "apparent differences" does mean statistically significant.

In Fig. 3B, there appears to be substantial muscle fiber remains in the decellularized scaffold (more than appears in other literature <https://www.sciencedirect.com/science/article/pii/S0142961208010685#fig1>). Have the authors stained for muscle membrane proteins (and/or cell membrane epitopes such as Galactose-alpha-1,3-galactose) that could lead to immune rejection of the ECM derived hydrogels? Alternatively, have the authors assessed activation of complement proteins to assess the immunogenicity of the ECM-derived gel? In the absence of this data, a further discussion of how this new hydrogel could eventually be translated to human patients, and how immune rejection of the gel could be mitigated would be helpful.

In line 271-272 the authors state that, "the proportion of CD206+ cells increased in all groups on day 3." Have the authors performed a statistical test to compare day 1 vs day3? That statistical test should be added to the manuscript. In fact, CD206+ does not appreciably increase in the soluble IL-4 treatment group between days 1 and 3. This makes sense given the slower/controlled cytokine release from the hydrogel, and is likely a strength of the hydrogel. This could be more clearly explained in the text.

In Fig. 5 composite extract was used to study its effects on BMDMs and L6 muscle cell recovery. While this tests that cytokines released from the hydrogel are bioactive, it does not allow study of the prolonged release profile from the hydrogel vs. soluble cytokine, or the effect of sequential release of IL4 then IGF-1. By separating L6 cells and BMDMs in the Transwell culture, this experiment also was not able to assess the effect of cell-cell interactions between the macrophages and muscle cells. Direct culture of the hydrogel, on top of the Transwell insert, and a true coculture in the bottom portion of the Transwell may show increased synergy.

In line 662 the authors conclude that, "the interpenetrating network structure that was formed between the microspheres and the hydrogel in the composites substantially enhanced the mechanical strength of the injectable mECM hydrogel." In order to conclusively state that the change in mechanical properties was due to the interpenetrating network, data for non-porous microspheres (or an otherwise non-interpenetrating network) would need to be provided. While such data seems outside the scope of this manuscript, line 662 should be adjusted to better reflect the data presented. To further support the role of the microspheres on the mechanical properties of the composite, perhaps data showing the effect of different concentrations of microspheres on

the mechanical properties of the composite could be shown, or discussed.

Minor comments

- It is unclear how the pore sizes were measured. The pores do not appear to be spherical. Are the reported pore sizes from the largest pore dimension or the smallest? The authors report an n=21 microspheres, were these microspheres all from the same lot or are the pore sizes consistent over multiple batches of microsphere synthesis?
- In Fig.2 H,J, At what time point after tweezer pinch were the microsphere diameters measured? In other words, how long does it take for the microspheres to regain their shape? Is IGF-1 released from the particle during the deformation/injection?
- Figure labels in the Fig.3 caption seem incorrect.
- It should be clearly stated in the text that Fig. S6 is day 1 data, and Fig. 4I-N is day 3.
- Letter labels need to be added to the parts of Fig. S6.
- The colors in Fig. 4G/H and in Fig. S5 are not consistent.
- There is no IL4 only group in fig 5. This makes it difficult to assess the effects of IL4 on muscle cells.
- Fig. 5 J-L, the figure caption lists n=3, but there are more data points than this in the graphs. How many images per sample were chosen for analysis and how were the images chosen? A more quantitative technique (other than quantification of select images from microscope slides) would further strengthen the data throughout Fig. 5. Perhaps plate reader based assays for quantification of fluorescence, or flow cytometry could be applied for some of the panels in Fig. 5.
- In line 503 the authors state that the quantity of muscle fibers is lower in the treatment groups than in normal muscle, but no quantification of muscle fibers is provided in the figures. (Perhaps the authors are referring to the size distribution of the fibers shown in Fig.7 rather than the 'quantity' of fibers?)
- Fig. 7H,I: How many rats were used (i.e. what is the n) the figure caption only says that 5 fields of view were quantified, but from how many rats?
- Fig. 8J, should clarify how many animals were used and how many images per animal were quantified.
- It would be easier for the readers to follow the discussion section, if a few references to the Figures containing the data being discussed were included.
- Line 694: based on fig. 9; it looks like the material components alone had a small effect and there was an increased effect with the cytokines added (blue vs black bar at 8wks Fig. 9F). These effects could maybe be considered additive but not synergistic. To truly assess the effect of growth factors and material, and then synergy between the two, the authors need to have a growth factor only group.
- Line 727: Should state that the composites can likely be applied to other tissues, as data from only one tissue was presented here.
- Section "Fabrication of the mECM@IL4+PM@IGF1 composites and release profile:" How much IGF-1 was added to the particles (i.e. what volume of 500ng/mg IGF-1 solution was used for 1.5mg of particles)? What is the approximate IGF-1 dose per particle and per ECM hydrogel?
- Section "The regulation of macrophage polarization in vitro:" What dose of IGF-1 was used?

Reviewer #3:

Remarks to the Author:

In this paper, the authors develop biomaterial system consisting of elastic porous microspheres mixed with ECM hydrogels as injectable composites with interleukin-4 (IL-4) and insulin-like growth factor-1 (IGF-1) dual-release functionality. The study is well designed, and the results obtain support the discussion and conclusions appropriately.

The authors describe the ability of the proposed system to provide good injectability and biocompatibility and to regulate the behavior of macrophages and myogenic cells following injection into muscle tissue. Therefore, the simultaneous regulation of macrophage and endogenous stem cell behavior may serve as a novel strategy to enhance in situ tissue regeneration. There are however some concerns regarding the novelty of this work, as a number of publications have reported the use of microporosity and injectability as a strategy for in situ tissue regrowth and regeneration and numerous articles advocate for promoting endogenous regeneration via immunomodulating biomaterials.

Overall I would recommend at least major revisions before accepting the manuscript for publications.

Overall comments regarding manuscript:

In the introduction a paragraph should be added to illustrate commercially available injectable materials and other approaches widely used for treatment of VML.

In the discussion section, although it's clear and well written the authors do not compare their results with existing works to discuss and highlight the differences and similarities of their outcomes. There is also lack of discussion about why there are some significant differences and no differences in certain markers with reference to time and experimental group.

In the conclusion section line 727.... "in vivo and in vivo" should be corrected.

Overall comments regarding experimental work:

The authors do not demonstrate a significant advantage compared to existing materials used by performing a comparative analysis.

The study lacks a long-term assessment to illustrate if it is immediate physical result or long-term volume maintenance is achieved.

The effect of optimal concentration of composite is also missing which can showcase in degradation studies if there is volume loss or swelling of material overtime.

The study does not mention the injectability window after mixing and loading in syringe, if any.

Also the uniform distribution and sedimentation of the microspheres post injection is not discussed.

The study does not document the general welfare of the animals, along with appearance of injection site and locomotive behavior of the animal. That can also add some context on visible swelling, redness, edema and abnormal color of the injection sites.

Local tissue response at the interface between the injected biomaterial and the native tissue should be incorporated as well. The number of lymphocytes, polymorphonuclear cells etc should be counted between field of view between interface and implant.

Evaluation of the thickness of the fibrotic capsule around the implant should also be included in the study. This can be estimated from the presence and thickness of a tightly packed layer of fibroblasts displaying increased eosin uptake.

In methodology section where images are used to quantify it should be presented with more details including exclusion and inclusion criteria set for quantification.

Some more cytokines should be included IL-23, IL-1 α , MCP-1, IL-12p70, IL-6, IL-27, IL-17A, IFN- β , and GM-CSF

Minor comments:

line 71.... can result in the strucutal and morphological collapse. "strucutal" spelling mistakes

line 727.... "in vivo and in vivo" should be corrected.

Reviewer #1 (Remarks to the Author):

This is a very interesting and innovative study on a timely and important topic in biomedical materials. The authors had constructed polymer microspheres with elastic characteristics, which filled the bottleneck problem of insufficient mechanical properties of porous microspheres. The developed mECM@IL4+PM@IGF1 multifunctional composites exhibited favorable injectability and biocompatibility, and achieved the regeneration of volumetric tibialis anterior muscle loss. In general, I think this work is interesting to readers and well-performed. I have some comments that need to be addressed by the authors.

Comment #1. The author said that synthetic polymers lack bioactivity, so how about the cytocompatibility of porous PLCL microspheres *in vitro*?

Response: Thank you for your professional comments. PLCL was FDA-approved biomaterial, and the biocompatibility of PLCL has been evaluated in our previous studies. For example, the multi-layer vascular scaffolds made of PLCL promoted the proliferation of endothelial cells and vascular smooth muscle cells *in vitro* [1], and PLCL vascular grafts achieved functional vascular regeneration *in vivo* at 12 months [2]. In order to evaluate biocompatibility of porous PLCL microspheres, different masses (0.5 mg, 1.5 mg, and 3.0 mg) of PLCL microspheres were co-cultured with L6 cells (**Fig. R1A, B**), L6 cells proliferated normally among all groups, and cell viability exhibited no significant difference compared with the control (TCP) group (**Fig. R1A**). And the L6 cells spread well in all groups after 3 days of co-culture (**Fig. R1B**). In addition, L6 cells could infiltrate into the interior of the microspheres (**Fig. R1C, D**), and the number of cells inside of the microspheres gradually increased over time. All these results confirmed the high cytocompatibility of porous PLCL microspheres.

Reference:

- [1] X. Yuan, W. Li, B. Yao, Z. Li, D. Kong, S. Huang, M. Zhu, Tri-Layered Vascular Grafts Guide Vascular Cells' Native-like Arrangement, *Polymers* 14(7) (2022).
- [2] M. Zhu, Y. Wu, W. Li, X. Dong, H. Chang, K. Wang, P. Wu, J. Zhang, G. Fan, L. Wang, J. Liu, H. Wang, D. Kong, Biodegradable and elastomeric vascular grafts enable vascular remodeling, *Biomaterials* 183 (2018) 306-318.

Fig. R1. The cytocompatibility of porous PLCL microspheres *in vitro*. A) Cell viability of L6 cells treated with microspheres of different masses on days 1, 2 and 3 (n=6 parallel samples per group). B) Images of L6 cells after co-culturing with microspheres on days 3. C) The cross-sectional pattern showing the cell position in microspheres. D) DAPI staining images showing cell distribution on days 1, 3, 5 and 7.

Comment #2. In Fig.3, the author needs to recheck the order of the figure legend number.

Response: We have revised the number order of the figure legend in the whole manuscript.

Comment #3. I suggest that the functionalization of microspheres with intracellular adhesion molecules rather than PDA/IGF-1 could be another direction. The authors could expand the discussion by referring to existing literature/ideas.

Response: Thanks for your suggestion. The introduction of hydroxyl, amino, polypeptide (RGD) on the surface of the elastic microspheres is expected to improve the efficiency of cell adhesion and drug delivery, which are worthy of further study [1]. We have added relevant discussions in line 802-804, please refer to the revised manuscript.

Reference:

[1] Q. Li, B. Chang, H. Dong, X. Liu, Functional microspheres for tissue regeneration, *Bioactive Materials* 25 (2023) 485-499.

Comment #4. More details on the DNA and ELISA assays should be included in the method section.

Response: We have added corresponding test methods in the ‘materials and methods’ section (line 915-923, line 955-960) of the revised manuscript.

Comment #5. Minor corrections I spotted so may as well mention:

1) In Fig.1, "injction" should be "injection", "bioactive" should be "bioactive".

2) Line 287, "IL--containing" should be "IL-4 containing".

3) In Fig.4A, "MC-CSF" should be "GM-CSF".

4) Line 354, "2.0±0.5 h" should be "2.0±0.5 hrs"

5) In Fig.5, "3 days" should be "3 Days".

Response: We were really sorry for our careless mistakes. The typo is revised in our resubmitted manuscript,

Reviewer #2 (Remarks to the Author):

In this work Li et al. describe the formulation of a porcine muscle ECM-derived hydrogel with embedded porous PLCL microspheres. By functionalizing the microspheres with IGF-1, and incorporating IL-4 in the ECM hydrogel, Li et al. demonstrate the controlled and sequential release of IL-4 then IGF-1 from the hydrogel composite. The authors nicely characterize the mechanical properties of the composite hydrogel, in a range that is appropriate for skeletal muscle, and demonstrate its injectability. Further, injection of the composite hydrogel, into a model of volumetric muscle loss in rats, resulted in improved muscle fiber histology, muscle perfusion, and functional innervation. In vitro studies were used to show the effect of the IL4/IGF1 composite hydrogels on macrophage polarization, and the effect of IGF1 on muscle cell growth.

Comment #1. While this is an interesting and versatile biomaterial for cytokine delivery, some additional data and discussion could be added to the manuscript to strengthen the significance and novelty of the work. ECM-based hydrogels as well as the immune polarizing and pro-regenerative effects of IL4 and IGF1 are well known in the field.

Response: Thank you very much for your positive evaluation on the significance and novelty of our work, and providing our many valuable suggestions. We have provided a point-to-point response to your comments.

Comment #2. Additional data showing how the embedded microspheres alter the mechanical properties of the composite (does microsphere pore size and/or sphere diameter change the properties of the composite? Can the density of microspheres be used to tune the mechanical properties of the composite?) would add to the versatility of the biomaterial and significance of the work.

Response: According to your suggestion, we have supplemented additional experiments about the effect of pore size and density of the embedded microspheres on the mechanical properties of composites in Fig. R2, 3 (Fig. S7, S5B-E in revised manuscript). As the porosity of porous microspheres increased (from 43.5±7.5 to 92.8±1.6%), the mechanical strength gradually decreased (Fig. R2A-E). Also, we

investigated the effect of porous microsphere density on the mechanical properties of composites (**Fig. R3A-D**). The mass ratio of mECM to microspheres is as follows: 30:1, 10:1, 5:1, and 3:1. As the mass ratio of mECM to PM decreased, the mechanical strength of the composites was significantly enhanced. PLCL microspheres played a major role in mechanical strength at a mass ratio of 5:1 and 3:1. The composites lost their gel-forming capabilities at a mass ratio of 3:1 (**Fig. R3D**). Therefore, it was difficult to construct stable composites when the mass ratio of mECM to PM was lower than 3:1.

Fig. R2. The effect of porosity of microspheres on the mechanical strength of composites. A) The porosity of porous PLCL microspheres at different gelatin concentrations (2.5%, 5.0%, 7.5% and 10.0%). B) Rheological tests of composites containing microspheres with different porosity, and the mass ratio of mECM to PM was 10:1 in all groups.

Fig. R3. The effect of porous microsphere density on the mechanical strength of composites. A-D) Rheological tests of composites with the different mass ratios of mECM to PM (30:1, 10:1, 5:1 and 3:1), and the gelatin concentration was 7.5% in all

groups.

Comment #3. While Li et al. show controlled cytokine release, the release profiles were not explored in a therapeutic context. (Is there a regenerative advantage of sequential cytokine release, does cytokine order matter? Is the biomaterial needed or are soluble cytokines sufficient? The *in vivo* data in figures 7-9 do not contain cytokine only groups.)

Response: Taking your advices. we have added individual growth factors treatment *in vivo*, including the mECM@IL4+PM group and mECM+PM@IGF1 group. The relevant results and discussion have been added to the revised manuscript in **Fig. 7-9**.

Muscle regeneration is a complex and sequential biological process that includes the inflammation phase, the repair and fibroplasia phase and the remodeling phase, and involves the participation of various immune cells, stem/progenitor cells, growth factors, and ECM components [1]. Recent studies have shown that, regulating the behavior of the infiltrated immune cells in early stage and muscle satellite cells in later stage through the orderly release of bioactive factors is crucial for muscle regeneration [1]. Biomaterials mainly provide a biomimetic microenvironment for cell proliferation, migration, and differentiation, which can serve as matrices to induce tissue regeneration, and in implants to support cell transplantation. Natural extracellular matrix as a typical biomaterial has been proven to promote VML regeneration in preclinical study [2], but its therapeutic effect is still limited. In order to improve the physiological VML regeneration, we constructed a functionalized composite named as mECM@IL4+PM@IGF1. In this composite, IL-4 was directly added into mECM hydrogel to regulate the immune microenvironment through rapid release, and IGF-1 with relatively slower release rate through polydopamine modification aimed to regulate the myogenic cells behavior. We demonstrated that the dual factor releases synergistically promoted macrophage polarization, and muscle satellite cells differentiation *in vitro*. We also confirmed the pro-regenerative effect of the dual factor composites was higher than that of the non-factor group and the single factor group *in vivo*.

Reference:

- [1] K.H. Nakayama, M. Shayan, N.F. Huang, Engineering Biomimetic Materials for Skeletal Muscle Repair and Regeneration, *Advanced Healthcare Materials* 8(5) (2019).
- [2] J. Dziki, S. Badylak, M. Yabroudi, B. Sicari, F. Ambrosio, K. Stearns, N. Turner, A. Wyse, M.L. Boninger, E.H.P. Brown, J.P. Rubin, An acellular biologic scaffold treatment for volumetric muscle loss: results of a 13-patient cohort study, *npj Regenerative Medicine* 1(1) (2016).

Comment #4. Lastly, the authors refer to synergy between IL4 and IGF1, but the

animals were not treated with the cytokines individually in the muscle studies (figures 7-9). So it is unclear if the effects of the two cytokines are both required for muscle regeneration, and if their combination is additive or synergistic. Biomaterials loaded with individual growth factors would add to the work. Control groups in figures 7-9, containing biomaterial without cytokines, suggest that the material itself has some beneficial effect on muscle regeneration. Additional discussion regarding the possible mechanism of the material itself would also strengthen the manuscript.

Response: According to your suggestion, we have added individual growth factors treatment group *in vivo* (**Fig. 7-9**), including mECM@IL4+PM group and mECM+PM@IGF1 group. We have comprehensively investigated the pro-regenerative effects of untreated, PM, mECM, mECM+PM, mECM@IL4+PM, mECM+PM@IGF1 and mECM@IL4+PM@IGF groups, and the relevant results and discussion have been added to the revised manuscript. Compared to the single factor treatment, the combination of IL-4 and IGF-1 showed a synergistic effect rather than an additive effect on promoting the muscle regeneration, vascularization, and neuralization.

Previous studies have revealed the pro-regenerative mechanism of extracellular matrix materials applied in muscle regeneration. Kaitlyn et al. proved that extracellular matrix scaffold activated mTOR/Rictor dependent Th2 signaling pathway to guide IL-4 dependent macrophage polarization and promoted damaged muscle regeneration by regulating the adaptive immune system [1]. In addition, extracellular matrix can also stimulate endogenous NK cells to produce chemokine Xcl-1 and recruit tDCs cells to promote the secretion of cytokines such as IL-4, IL-10, and IL-33 in the defect sites, thereby improving the efficiency of myofibers regeneration [2]. We have added relevant mechanism descriptions in the discussion section (line 773-782), please refer to the revised manuscript.

Reference:

- [1] K. Sadtler, K. Estrellas, B.W. Allen, M.T. Wolf, H. Fan. Developing a pro-regenerative biomaterial scaffold microenvironment requires T helper 2 cells, *Science* 352(6283) (2016) 366-370.
- [2] R. Lokwani, A. Josyula, T.B. Ngo, S. DeStefano. Pro-regenerative biomaterials recruit immunoregulatory dendritic cells after traumatic injury, *Nature Materials* (2023).

Comments

Comment #5. The authors should take greater care when describing comparisons between groups. Throughout the manuscript (and especially in the paragraphs starting on lines 266, and 456), the authors describe comparisons that are ‘apparent,’ or ‘noticeable;’ sometimes these differences are statistically significant and other times they are not. It would help the reader to critically follow the data if differences that are significant are consistently described as ‘significant,’ and those that are not statistically

significant are explicitly described as ‘not statistically significant.’

Examples: The authors should clarify that the difference between the control and ECM+PM group described in line 274 is not statistically significant. In lines 280–282 the authors state that the difference between the pure IL-4 group and the mECM@IL4+PM was ‘noticeable,’ however the data in Fig. 4H suggests that this difference is not statistically significant.

In line 562 “apparent differences” does mean statistically significant.

Response: Thank you for bringing us the valuable suggestion. According to the reviewer’s suggestions, we have unified the description of significant differences in the whole manuscript, please refer to the revised manuscript.

Comment #6. In lines 492 – 495, it should be made clear that the differences between the hydrogel+IGF1+IL4 group vs. the ECM only and ECM+PM groups are not statistically significant. (Alternatively, only reference the significant differences in the text).

Response: According to the reviewer’s suggestion, we have simplified the relevant description in the revised manuscript (ling542-546).

Comment #7. In Fig. 3B, there appears to be substantial muscle fiber remains in the decellularized scaffold (more than appears in other literature <https://www.sciencedirect.com/science/article/pii/S0142961208010685#fig1>). Have the authors stained for muscle membrane proteins (and/or cell membrane epitopes such as Galactose-alpha-1,3-galactose) that could lead to immune rejection of the ECM derived hydrogels? Alternatively, have the authors assessed activation of compliment proteins to assess the immunogenicity of the ECM-derived gel? In the absence of this data, a further discussion of how this new hydrogel could eventually be translated to human patients, and how immune rejection of the gel could be mitigated would be helpful.

Response: Thank you very much for your in-depth comments. As you mentioned, the muscle fiber morphology in this literature [1] was inconsistent with our results in **Fig. 3B**. This is mainly due to the different decellularization processes. The size of muscle tissue, the type of detergent, and the time of decellularization all affect the ultrastructure of muscle extracellular matrix. In this literature, the authors cut the muscle tissues into thin slices with a thickness of less than 500 μm , and treated with 1% Triton X-100 for 5 days, which easily lead to changes in the muscle internal ultrastructure. Whilst, we cut muscle tissue into small pieces with a thickness of approximately 5 mm in **Fig.3B**, and treated with 1% sodium dodecyl sulfate (SDS) for 3 days. As shown in **Fig. 3B**,

this method can maintain the tissue ultrastructure while removing most of the cell nuclei and DNA (Fig. 3B, F&G). α -gal is the main target antigen that causes hyperacute immune rejection of animal-derived biomaterials. The activity of α -gal decreased significantly from 2689.9 ± 202.7 U/g to 287.0 ± 148.9 U/g after decellularization, which was detected using the α -gal kit in Fig. R4 (Fig. 3H in revised manuscript). Moreover, western-blot showed that the α -gal was not detectable after decellularization (Fig. R5). We refined the decellularization method and added the α -gal test method (line 853-855) and results (line 190-191) to the revised manuscript.

Reference:

[1] M.M. Stern, R.L. Myers, N. Hammam, K.A. Stern. The influence of extracellular matrix derived from skeletal muscle tissue on the proliferation and differentiation of myogenic progenitor cells ex vivo, *Biomaterials* 30(12) (2009) 2393-2399.

Fig. R4. The activity detection of α -galactosidase (α -gal) before and after muscle decellularization (n=3 samples per group)

Fig. R5. Western-blot detected the α -galactosidase (α -gal) before and after muscle decellularization (n=3 samples per group)

Comment #8. In line 271-272 the authors state that, “the proportion of CD206+ cells increased in all groups on day 3.” Have the authors performed a statistical test to compare day 1 vs day3? That statistical test should be added to the manuscript. In fact, CD206+ does not appreciably increase in the soluble IL-4 treatment group between days 1 and 3. This makes sense given the slower/controlled cytokine release from the hydrogel, and is likely a strength of the hydrogel. This could be more clearly explained in the text.

Response: Thanks for the specific suggestion. We have added the statistical analysis

and relevant description to compare day 1 and day 3, as showed in **Fig. R6 (Fig. S5D in revised manuscript)**.

Fig. R6. The change trend of the proportion of CD206⁺ cells from day 1 to day 3 (n=3 samples per group).

Comment #9. In Fig. 5 composite extract was used to study its effects on BMDMs and L6 muscle cell recovery. While this tests that cytokines released from the hydrogel are bioactive, it does not allow study of the prolonged release profile from the hydrogel vs. soluble cytokine, or the effect of sequential release of IL4 then IGF-1. By separating L6 cells and BMDMs in the Transwell culture, this experiment also was not able to assess the effect of cell-cell interactions between the macrophages and muscle cells. Direct culture of the hydrogel, on top of the Transwell insert, and a true coculture in the bottom portion of the Transwell may show increased synergy.

Response: According to your suggestion, we re-conducted the cell co-culture experiment, as showed in **Fig. R7 (Fig.5I-L in revised manuscript)**. BMDMs were directly inoculated into the lower chamber and co-cultured with CTX-injured satellite cells, and the composites were directly placed on the upper chamber to comprehensively investigate the interaction among BMDMs, CTX-injured satellite cells and composites. The mECM@IL4+PM@IGF1 composites not only promoted BMDMs polarization into CD206⁺ phenotype, but also further improved the muscle satellite cells differentiation cooperated with CD206⁺ BMDMs. The modified methods were added in the materials and methods section, and the results were included in **Fig.5I-L** in revised manuscript.

Fig. R7. mECM@IL4+PM@IGF1 composites and BMDMs modulated the muscle satellite cells differentiation. A) Representative images of phalloidin co-staining with CD206 and desmin antibody showing the differentiation-promoting effect of BMDMs and composites on injured muscle satellite cells. Scale bar: 50 μm. B-D) Quantitative statistics of the myotube length, the myotube fusion area, and the nuclei number per myotube (n=9 myotubes, from 3 parallel samples). E) Quantitative statistics of CD206 fluorescence intensity (n=3 parallel samples per group).

Comment #10. In line 662 the authors conclude that, “the interpenetrating network structure that was formed between the microspheres and the hydrogel in the composites substantially enhanced the mechanical strength of the injectable mECM hydrogel.” In order to conclusively state that the change in mechanical properties was due to the interpenetrating network, data for non-porous microspheres (or an otherwise non-interpenetrating network) would need to be provided. While such data seems outside the scope of this manuscript, line 662 should be adjusted to better reflect the data presented. To further support the role of the microspheres on the mechanical properties of the composite, perhaps data showing the effect of different concentrations of microspheres on the mechanical properties of the composite could be shown, or discussed.

Response: Thanks for in-depth comments, we have modified the inappropriate description, the mechanical change of composites was mainly related to the PLCL microspheres. Indeed, it is difficult for us to evaluate the mechanical contribution of interpenetrating structures. Based on your suggestion, we investigated the effects of porous microsphere density on mechanical properties of composites, the relevant result was shown in Fig. R3 (comment #2).

Minor comments

Comment #11. It is unclear how the pore sizes were measured. The pores do not appear to be spherical. Are the reported pore sizes from the largest pore dimension or the smallest? The authors report an n=21 microspheres, were these microspheres all from the same lot or are the pore sizes consistent over multiple batches of microsphere synthesis?

Response: Sorry for the confusion. We measured the relatively average pore size of porous microspheres, rather than the maximum or minimum pore size. The measurement method has been added to the materials and methods section. Briefly, the pore area ‘S’ of porous microspheres in SEM images was calculated using image pro-plus software, and the relative pore sizes ‘d’ were calculated by the formula $S = \pi(d/2)^2$, as shown in **Fig. R8**. The method was added in revised manuscript (line 882-884). Furthermore, the 21 microspheres were from at least 3 batches of samples.

Fig. R8. The calculation formula of pore sizes of microspheres.

Comment #12. In Fig.2 H, J, at what time point after tweezer pinch were the microsphere diameters measured? In other words, how long does it take for the microspheres to regain their shape? Is IGF-1 released from the particle during the deformation/injection?

Response: Thank you for your detailed question. Porous PLCL microspheres can achieve rapid morphology recovery within 3 seconds after pressing, as exhibited in **supplementary video 1, table 1 and table 2**. The results showed that the larger diameter and higher porosity of microspheres displayed the longer morphology recovery time (**table 1&table 2**). To ensure the full restoration of microspheres original morphology, all the images were collected 30 seconds after pressing.

Table 1: Recovery time of porous PLCL microspheres with different needle sizes.[†]

Different needle sizes [†]	Diameter (μm) [†] (Mean ± SD) [†]	Recovery time (s) [†] (Mean ± SD) [†]
18 G [†]	818.1 ± 88.3 [†]	2.13 ± 0.22 [†]
20 G [†]	607.3 ± 73.7 [†]	1.91 ± 0.16 [†]
21 G [†]	544.4 ± 42.5 [†]	1.78 ± 0.19 [†]
26 G [†]	360.0 ± 28.6 [†]	1.51 ± 0.27 [†]

Table 2: Recovery time of porous PLCL microspheres with different gelatin concentrations.

Different gelatin concentrations (%)	Porosity (%) (Mean \pm SD)	Recovery time (s) (Mean \pm SD)
2.5	43.5 \pm 7.5	1.07 \pm 0.23
5.0	76.6 \pm 5.1	1.45 \pm 0.40
7.5	83.7 \pm 2.8	1.42 \pm 0.42
10.0	92.8 \pm 1.6	2.89 \pm 0.65

In addition, we evaluated the release behavior of IGF-1 from elastic porous microspheres (PM@IGF-1) after injection and compression in **Fig. R9 (Fig. S8 in revised manuscript)**. The IGF-1 release amount in injection group (samples collected post-injection 24 hrs) exhibited no significant difference compared with the non-injection group (**Fig. R9A, B**). In addition, the IGF-1 release rate in compression group (compressed with a 20 g weight for 30 s) showed no significant difference compared with the non-compression group at different time points (0, 12 hrs, 24 hrs, 72 hrs, 120 hrs and 168 hrs) (**Fig. R9C, D**). The above results indicated that the external forces have no obvious effect on the release behavior of IGF-1.

Fig. R9. IGF-1 release behavior after injection and deformation. A) Schematic diagram of injection in vitro. B) The release amount of IGF-1 on day 1 after injection. C) Pattern diagram of parallel plate compression testing. D) Release profiles of IGF-1 under compression or non-compression.

Comment #13. Figure labels in the Fig.3 caption seem incorrect.

Response: Sorry for our careless mistakes. We have corrected it in revised manuscript (line 229-241).

Comment #14. It should be clearly stated in the text that Fig. S6 is day 1 data, and Fig. 4I-N is day 3.

Response: We have taken the reviewer's opinion and enlarged the annotation, please refer to the revised manuscript (line293-295).

Comment #15. Letter labels need to be added to the parts of Fig. S6.

Response: Thanks for your reminding. We have added the letter labels in Fig. S6.

Comment #16. The colors in Fig. 4G/H and in Fig. S5 are not consistent.

Response: We have already unified the colors.

Comment #17. There is no IL4 only group in fig 5. This makes it difficult to assess the effects of IL4 on muscle cells.

Response: Thanks for the reviewer's advice. We have evaluated the effects of IL-4 and IGF-1 on the proliferation of myogenic cells in vitro, and IL-4 displayed no obvious effects on myogenic cells in Fig. R10 (Fig. S11 in revised manuscript).

Fig.R10. The synergistic effects of IL-4 and IGF-1 on the proliferation of L6 cells on 1 and 3 days (n=6 parallel samples per group).

Comment #18. Fig. 5 J-L, the figure caption lists n=3, but there are more data points than this in the graphs. How many images per sample were chosen for analysis and how were the images chosen? A more quantitative technique (other than quantification of select images from microscope slides) would further strengthen the data throughout Fig. 5. Perhaps plate reader based assays for quantification of fluorescence, or flow cytometry could be applied for some of the panels in Fig. 5.

Response: Sorry for the confusion. Based on your suggestion, we have supplemented the experiment and replaced the images in Fig.R7 (comment#9). The figure caption lists 'n=9' indicated 9 myotubes were randomly selected from 3 parallel samples (Fig.R7B-D). And 'n=3' indicated the number of CD206⁺ cells were counted from 3 parallel samples in Fig.R7E. In addition, microplate reader was used to detect cell viability in Fig.5E, R10, and fluorescence microplate quantitative technique was applied to detect calcium ion fluorescence intensity in Fig.5G. Please refer to the revised manuscript.

Comment #19. In line 503 the authors state that the quantity of muscle fibers is lower in the treatment groups than in normal muscle, but no quantification of muscle fibers is provided in the figures. (Perhaps the authors are referring to the size distribution of the fibers shown in Fig.7 rather than the ‘quantity’ of fibers?)

Response: Thanks for the reviewer’s careful reading. The quantity of myofibers was determined through desmin fluorescence images in Fig.7E. According to your suggestion, we supplemented the number of myofibers in Fig. R11. However, the effect of muscle regeneration could not be accurately evaluated by the quantity of myofibers, due to the wide distribution and small area of muscle fibers at 2 weeks. Indeed, the distribution of the cross-sectional areas (CSA) of myofibers can better reflect muscle regeneration (Fig. 5I-T), which have been applied in other studies [1, 2].

Reference:

[1] Y. Jin, E.J. Jeon, S. Jeong, S. Choi, S.W. Cho. Reconstruction of Muscle Fascicle-Like Tissues by Anisotropic 3D Patterning, *Advanced Functional Materials* 31(25) (2021).

[2] Y. Jin, D. Shahriari, E.J. Jeon, S.W. Cho. Functional Skeletal Muscle Regeneration with Thermally Drawn Porous Fibers and Reprogrammed Muscle Progenitors for Volumetric Muscle Injury, *Advanced Materials* 33(14) (2021).

Fig. R11. Quantitative statistics of the number of myofibers. (n=3 samples per group, 2 random fields per sample).

Comment #20. Fig. 7H, I: How many rats were used (i.e. what is the n) the figure caption only says that 5 fields of view were quantified, but from how many rats?

Response: Sorry for missing quantitative data of one field of view, and we included it in Fig.7H. In fact, 3 rats per group were used *in vivo* experiments, and 2 random fields per sample were quantified. We have revised the corresponding figure caption (line 573-575).

Comment #21. Fig. 8J, should clarify how many animals were used and how many

images per animal were quantified.

Response: 3 rats per group were used *in vivo* experiments, and 2 fields of view per sample were quantified (line 634-635).

Comment #22. It would be easier for the readers to follow the discussion section, if a few references to the Figures containing the data being discussed were included.

Response: Thank you for your comments. We have added references to the figures containing the data being discussed. Please refer to the revised manuscript.

Comment #23. Line 694: based on fig. 9; it looks like the material components alone had a small effect and there was an increased effect with the cytokines added (blue vs black bar at 8wks Fig. 9F). These effects could maybe be considered additive but not synergistic. To truly assess the effect of growth factors and material, and then synergy between the two, the authors need to have a growth factor only group).

Response: We appreciated your suggestions and added two individual factor groups (mECM@IL4+PM and mECM+PM@IGF1) in *in vivo* experiment (Fig.7-9). Compared with the mECM@IL4+PM group or mECM+PM@IGF1 group, the mECM@IL4+PM@IGF1 group showed a synergistic effect rather than an additive effect on promoting the myofiber regeneration, vascularization, and neuralization. Please refer to the revised manuscript.

Comment #24. Line 727: Should state that the composites can likely be applied to other tissues, as data from only one tissue was presented here.

Response: We have polished the expression in conclusion section in the revised manuscript (line 811-813).

Comment #25. Section “Fabrication of the mECM@IL4+PM@IGF1 composites and release profile:” How much IGF-1 was added to the particles (i.e. what volume of 500ng/mg IGF-1 solution was used for 1.5mg of particles)? What is the approximate IGF-1 dose per particle and per ECM hydrogel?

Response: Sorry for not making this clear. 500 μ L IGF-1 solution (500 ng/mL stated in labeling information) were added to the PDA-modified porous PLCL microspheres (1.5 mg). Furthermore, we measured the IGF-1 dose loading in microspheres by ELISA kit. Specifically, to load IGF-1 to the microspheres, 3 mg PDA-modified microspheres were incubated with 1 mL IGF-1 solution for 24 hrs at 4°C (Fig. R12). Afterwards, to measure the loading amount of IGF-1, the concentration of IGF-1 in unmodified solution and modified solution were quantified respectively by ELISA kit. Based on the ELISA results (Fig. R12), the loading amount of IGF-1 in 3 mg microspheres was 145.8 ± 19.2 ng. The number of porous microspheres per milligram was approximately

49±7, it could be calculated that 1.02±0.18 ng IGF-1 was loaded in per microsphere.

Fig. R12. A) Schematic illustration of the loading of IGF-1 to the PDA-modified microspheres. The supernatant after modification was collected for IGF-1 quantification. B) Bar diagram of the IGF-1 concentration in unmodified solution group and modified solution group.

Comment #26. Section “The regulation of macrophage polarization in vitro:” What dose of IGF-1 was used?

Response: For macrophage polarization experiment, 200 μ L composites (mECM:PM=10:1) were added to the upper chamber of transwell, which was contained approximately 0.6 mg microspheres loading with 29.2±3.8 ng IGF-1. It can be calculated from the above test result (comment#25).

Reviewer #3 (Remarks to the Author):

In this paper, the authors develop biomaterial system consisting of elastic porous microspheres mixed with ECM hydrogels as injectable composites with interleukin-4 (IL-4) and insulin-like growth factor-1 (IGF-1) dual-release functionality. The study is well designed, and the results obtain support the discussion and conclusions appropriately.

The authors describe the ability of the proposed system to provide good injectability and biocompatibility and to regulate the behavior of macrophages and myogenic cells following injection into muscle tissue. Therefore, the simultaneous regulation of macrophage and endogenous stem cell behavior may serve as a novel strategy to enhance in situ tissue regeneration. There are however some concerns regarding the novelty of this work, as a number of publications have reported the use of microporosity and injectability as a strategy for in situ tissue regrowth and regeneration and numerous articles advocate for promoting endogenous regeneration via immunomodulating biomaterials.

Overall I would recommend at least major revisions before accepting the manuscript for publications.

Response: Thank you for your feedback and raising the concern regarding the novelty of our study. Although several publications have reported the use of microporosity and injectability for in situ tissue regrowth and regeneration via immunomodulating biomaterials, the innovation of our study lies in material design and further functional modification, which is as followed:

1) Numerous studies have indicated that synthetic polymer microspheres can serve as vehicles for the loading of live cells or bioactive factors to promote tissue regeneration. But most porous microspheres prepared from synthetic polymers lack elasticity, and thus easily undergo deformation under compression and wall shear stress during injection, which can result in the structural and morphological collapse of microspheres. The elastic porous PLCL microspheres prepared in our research can compensate for the above shortcomings.

2) Secondly, we constructed an injectable composite system composed of elastic porous PLCL microspheres and ECM hydrogel for in situ tissue regeneration. The injectable composites that combine the complementary advantages gained from ECM hydrogels and microspheres, ECM hydrogels provided biocompatibility and polymer microspheres possessed the necessary mechanical strength and porous structure to guide tissue regeneration.

3) Finally, in order to further improve the pro-regenerative effect of the composite system, two bioactive factors were loaded into the composites. Sustained release of interleukin (IL)-4 (from ECM hydrogels) and IGF-1 (from microspheres) could synergistically regulate the behavior of macrophages and myogenic cells, thereby promoting myofiber formation, vascularization, neurogenesis, and recovery of muscle function.

Overall comments regarding manuscript:

Comment #1. In the introduction a paragraph should be added to illustrate commercially available injectable materials and other approaches widely used for treatment of VML.

Response: Thanks for the reviewer's advice. According to our manuscript, we considered it was suitable to be added to the discussion section, please see the revised manuscript.

Autologous local muscle flap transplantation is the 'gold standard' for the clinical treatment of VML. However, the suitable autografts are limited, and the flap transplantation can lead to complications such as donor site morbidity and infection [1]. Up to now, there is no clinically available products for VML treatment. In a clinical study, Badylak, etc. used xenogeneic acellular materials (MtriStem, BioDesign, and XenMatrix) to repair VML, which could increase muscle mass and functional recovery, exhibiting a certain pro-regenerative effect [2]. However, these types of products are mainly ECM scaffolds derived from porcine dermis, bladder, or small intestinal

submucosal, which are different from muscle ECM and lack of tissue specificity. These materials have no injectability (line 747-755).

Indeed, commercial injectable materials can achieve local tissue repair through minimally invasive procedures, which effectively overcome the limitations of traditional open surgery. For instance, the injectable hyaluronic acid Restylane™, Sunmax collagen implant I™ and flowable resin 3M™ Filtek™ were widely used in facial filling and dental restoration. However, these injectable materials were not designed and applied for VML regeneration.

Reference:

- [1] K.H. Nakayama, M. Shayan, N.F. Huang, Engineering Biomimetic Materials for Skeletal Muscle Repair and Regeneration, *Advanced Healthcare Materials* 8(5) (2019).
- [2] J. Dziki, S. Badylak, M. Yabroudi, B. Sicari, F. Ambrosio, K. Stearns, N. Turner, A. Wyse, M.L. Boninger, E.H.P. Brown, J.P. Rubin, An acellular biologic scaffold treatment for volumetric muscle loss: results of a 13-patient cohort study, *npj Regenerative Medicine* 1(1) (2016).

Comment #2. In the discussion section, although it's clear and well written the authors do not compare their results with existing works to discuss and highlight the differences and similarities of their outcomes. There is also lack of discussion about why there are some significant differences and no differences in certain markers with reference to time and experimental group.

Response: Thanks for your in-depth advice. Up to now, there have been no similar reports on the use of injectable mECM and elastic porous microsphere composites for tissue repair, including VML repair. Although many materials were developed for muscle repair, most of them are made of single synthetic polymers or natural materials, which have their own advantages and weakness. For instances, methacrylic acid (MAA) based hydrogels or ECM hydrogels could regulate the behavior of macrophages, and thus promote vascularization and neuralization in muscle regeneration [1-3]. However, these hydrogels with weak mechanical strength were unable to offer physical support. Synthetic polymers PLLA, PLGA, and PCL are commonly used in musculoskeletal tissue engineering [4], Chen et al. constructed injectable cell-laden PLGA microspheres to promote VML repair [5, 6], but the anti-deformation and injectable properties of porous PLGA microspheres have not been verified.

Compared with these materials (such as hydrogels or polymer materials), our composites not only have high biocompatibility, but also have porous structure to provide physical support for cell and tissue ingrowth. On the basis of composites, materials loaded with bioactive substances were able to regulate the immune cells and myogenic cells, which further synergistically promote myofiber regeneration, vascularization, and neuralization.

The regeneration process of VML is complex and involves a multicellular interaction network. In addition to macrophages and muscle satellite cells, immune cells (T, B, NK cells), fibroblasts, adipocytes, etc. can affect the process of tissue regeneration, which can lead to no significant difference in regeneration effects between individual groups. Additionally, our statistical methods are as followed: Single comparisons were performed using an unpaired Student's t test. Multiple comparisons were performed using Tukey's post hoc test and one-way analysis of variance (ANOVA). Multivariant comparisons were performed using two-way ANOVA with Tukey's post hoc test. Different statistical methods can also affect the significance of the difference among groups.

Reference:

- [1] M.M. Carleton, M. Locke, M.V. Sefton, Methacrylic acid-based hydrogels enhance skeletal muscle regeneration after volumetric muscle loss in mice, *Biomaterials* 275 (2021).
- [2] M.M. Carleton, M.V. Sefton, Injectable and degradable methacrylic acid hydrogel alters macrophage response in skeletal muscle, *Biomaterials* 223 (2019).
- [3] Y. Jin, E.J. Jeon, S. Jeong, S. Min, N. Choi, S.W. Cho. Reconstruction of muscle fascicle-like tissues by anisotropic 3D patterning, *Advanced Functional Materials* 31(25) (2021).
- [4] B.T. Corona, S.M. Greising, Challenges to acellular biological scaffold mediated skeletal muscle tissue regeneration, *Biomaterials* 104 (2016) 238-246.
- [5] Y. Wang, R.K. Kankala, Y.-Y. Cai, Y.S. Zhang, A.-Z. Chen. Minimally invasive co-injection of modular micro-muscular and micro-vascular tissues improves in situ skeletal muscle regeneration, *Biomaterials* 277 (2021).
- [6] R.K. Kankala, J. Zhao, Y.S. Zhang, A.Z. Chen. Highly porous microcarriers for minimally invasive in situ skeletal muscle cell delivery, *Small* 15(25) (2019).

Comment #3. In the conclusion section line 727.... "in vivo and in vivo" should be corrected.

Response: Thanks for the reviewer's careful reading, we have corrected it in revised manuscript in line 811.

Overall comments regarding experimental work:

Comment #4. The authors do not demonstrate a significant advantage compared to existing materials used by performing a comparative analysis.

Response: Thanks for the constructive comments. According to reviewer's suggestion, we compared the composites with existing materials for tissue repair in discussion section (line 678-706). Please refer to the revised manuscript.

Numerous studies have shown that ECM hydrogels possess high biocompatibility

but lack mechanical strength and degrade rapidly. Synthetic polymer porous microspheres can guide tissue regeneration, but their degradation byproducts easily trigger a local inflammatory response, limiting tissue regeneration. We overcame these shortcomings by combining the advantageous mechanophysical and cell supporting properties of porous microspheres with the injectability and biocompatibility of mECM hydrogels.

Synthetic polymeric porous microspheres are usually applied as injectable materials to repair irregular tissue defects. Synthetic polymers, such as PLGA, PCL, PLA, etc., have been frequently employed to prepare porous microspheres, but the resultant microspheres are prone to collapse or deformation after injection. These deficiencies are rarely reported. However, PLCL materials have inherent flexibility, elasticity and tensile properties that out-perform PLGA and PCL. In this study, we selected PLCL to construct elastic porous microspheres to allow for the recovery of original shape after pressing or injection. PLGA and PCL microspheres collapsed and broke or deformed after the compression tests, and thus were ruled out as candidate microsphere materials.

So far, no injectable composites composed of extracellular matrix hydrogel and elastic porous polymer microspheres developed for muscle regeneration has been reported. Some studies have constructed composite systems [1, 2], these composites were mainly designed for drug delivery, and did not take into account the effects of pore structure, elasticity, material compatibility, and complementarity of synthetic polymer and ECM on in vitro injection and in vivo tissue regeneration.

Reference:

- [1] X. Ji, H. Shao, X. Li, M.W. Ullah, G. Yang, Y. Zhang, Injectable immunomodulation-based porous chitosan microspheres/HPCH hydrogel composites as a controlled drug delivery system for osteochondral regeneration, *Biomaterials* 285 (2022).
- [2] J. Li, H. Jiang, Z. Lv, Z. Sun, C. Cheng, G. Tan, Q. Jiang, D. Shi, Articular fibrocartilage-targeted therapy by microtubule stabilization, *Science Advances* 8(46) (2022).

Comment #5. The study lacks a long-term assessment to illustrate if it is immediate physical result or long-term volume maintenance is achieved.

Response: Thanks for the constructive comments. According to reviewer's suggestion, we supplemented long-term animal experiments to evaluate the pro-regenerative effect of composites in **Fig. R13**.

The muscle volume of the injured side was still smaller than that of the normal side, and there were still a few visible microspheres observed in the PM, mECM+PM and mECM@IL4+PM@IGF1 groups at 16 weeks (**Fig.R13A**). The mass ratio in the

mECM@IL4+PM@IGF1 group was still higher than that in other groups, and showing a significant difference with untreated group (**Fig.R13B**). A very small amount of undegraded porous microspheres observed in PM, mECM+PM and mECM@IL4+PM@IGF1 groups, and a highest density of myofibers was distributed within the defect site in mECM@IL4+PM@IGF1 group, compared with other groups (**Fig.R13C**). In addition, at 16 weeks, the density of myofibers increased compared with 8 weeks in all groups. The mECM@IL4+PM@IGF1 group produced the highest density of thick myofibers compared with other groups, and it was closer to the normal muscle tissue (**Fig.R13D**), the desmin statistics also confirmed this result (**Fig.R13E**). The number of capillaries in the mECM@IL4+PM@IGF1 group was lower than the untreated, PM, mECM and mECM+PM groups, and closer to the normal group (**Fig.R13F, G**). Furthermore, the number of α -SMA⁺ mature blood capillaries in the mECM@IL4+PM@IGF1 group was also lower than that in the untreated, PM, mECM, and mECM+PM groups, and closer to the normal group (**Fig.R13H, I**). The maximum amplitude of the CAMP in the mECM@IL4+PM@IGF1 group was higher than other groups, and exhibited a significant difference from that of untreated, PM and mECM groups (**Fig.R13J, K**). NF-09⁺ expression in the mECM@IL4+PM@IGF1 group was also significantly higher than that of the untreated and PM groups, and similar to normal muscle (**Fig.R13L, M**).

Fig. R13. mECM@IL4+PM@IGF1 composites promote myofiber formation, vascularization and neuralization in VML model of rats at 16 weeks. A) Macroscopic images of muscle regeneration at 16 weeks post-treatment. B) The muscle mass ratio of the injured side/normal side (n=3 samples per group). C, D) H&E staining and desmin immunofluorescence images showing muscle regeneration. Scale bar in C for low magnification: 500 μm , high magnification: 100 μm . Scale bar in D: 50 μm . E) Statistical analysis of desmin fluorescence intensity in D (n=3 samples per group, 2 random fields per sample). F) H&E staining showing functional capillary formation. Black triangular arrows: capillary. Scale bar: 50 μm . G) The number of capillaries in F (n=3 samples per group, 2 random fields per sample). H) α -SMA immunofluorescence images showing mature capillary formation. Scale bar: 50 μm . I) Statistical results of α -SMA⁺ mature capillaries (n=3 samples per group, 2 random fields per sample). J)

Representative patterns of CMAP in different groups. K) Quantitative analysis of CMAP maximum amplitudes (n=3 samples per group). L) NF-09 immunofluorescence staining images showing neuralization. Scale bar: 50 μm . M) The statistical results of NF-09⁺ fluorescence intensity (n=3 samples per group, 2 random fields per sample).

Comment #6. The effect of optimal concentration of composite is also missing which can showcase in degradation studies if there is volume loss or swelling of material overtime.

Response: Thank you for your opinion. We supplemented the in vitro and in vivo experiments to confirm the optimal concentration of composites. Microspheres degraded over time, and only a small number of microspheres remained at 8 weeks, and no obvious volume swelling of microspheres were observed (**Fig. 7C**). Furthermore, we investigated the effect of different mECM to PM mass ratio on the injectability, stability, and mechanics of composites. Firstly, we prepared four composites with different mass ratios, including 30:1, 10:1, 5:1, and 3:1. All composites exhibited good injectability and dispersibility when injected immediately after preparation (**supplementary videos 4-7**). However, microspheres will settle down within 5 mins in 3:1 group, while in other groups, microspheres can be uniformly distributed in the mECM hydrogels within 48 hrs. This indicated that the high mass ratio of polymer porous microspheres was not conducive to the formation of a stable composite system in Fig.R14 (**Fig. S5 in revised manuscript**). As the mass ratio of mECM to PM decreased, the mechanical strength of the composites was significantly enhanced. PLCL microspheres played a major role in mechanical strength at a mass ratio of 5:1 and 3:1. The composites lost their gel-forming capabilities at a mass ratio of 3:1 (**Fig. R14E**). Therefore, it was difficult to construct stable composites when the mass ratio of mECM to PM was lower than 3:1.

Fig. R14. Stability and mechanical testing of composites with different mass ratios. A) Sedimentation testing of mECM and porous microsphere composites at different time points. B-E) Rheological tests of composites with different mass ratios.

Furthermore, we investigated the pro-regenerative effects of three injectable composite systems with different mass ratios (30:1, 10:1, and 5:1) using a rat VML model. After 2 weeks of implantation, the thickness of neo-muscles in 10:1 group was significantly thicker than that of the 30:1 and 5:1 groups in **Fig. R15 (Fig. S6 in revised manuscript)**. Excessive PLCL microspheres will accumulate in the defect areas in 5:1 group, inhibiting myofibers formation (**Fig. R15A, B**). The desmin expression in the 5:1 group was significantly lower than that of the 30:1 and 10:1 groups, which was confirmed by quantitative results (**Fig. R15C, D**). These indicated that the composites with a mECM to PM mass ratio of 10:1 was the optimal concentration for muscle regeneration.

In summary, the composites with mass ratio of 10:1 displayed high injectability and stability in vitro, as well as the pro-regenerative capability in vivo, which is optimal ratio for the application.

Fig. R15. mECM@IL4+PM@IGF1 composites with different mass ratio (30:1, 10:1, 5:1) promote muscle regeneration in VML model of rats at 2 weeks. A) H&E staining images showing muscle regeneration. Scale bar: 1 mm. B) The statistical results of neomuscle thickness (n=3 samples per group). C) Desmin immunofluorescence images showing neo-muscle regeneration. Scale bar: 50 µm. D) Statistical analysis of desmin fluorescence intensity in C (n=3 samples per group).

Comment #7. The study does not mention the injectability window after mixing and loading in syringe, if any. Also the uniform distribution and sedimentation of the microspheres post injection is not discussed.

Response: Thank you for your professional comments. The digested extracellular matrix can be stored at 4 °C for 1 month without affecting its gel-forming capability. Porous PLCL microspheres can also be stored for a long time at 4 °C after freeze-drying. However, we recommended immediate injection of composites after mixing to avoid degradation and inactivation of dual bioactive factors. Microspheres distributed

uniformly in the mECM hydrogel in 30:1, 10:1, and 5:1 group, but showed rapid sedimentation and aggregation in 3:1 group. Actually, the degradation rate of mECM is too fast, making it difficult to provide long-term spatial support for host cells and tissues ingrowth. The porous microspheres mainly settled at the bottom of the damaged area in PM group, while the mECM enabled the uniform distribution of porous microspheres in the damaged area in composites group, as showed in H&E staining (**Fig. 7C, S14C**). Microspheres would move towards the center collectively accompanied by the ingrowth of neo-muscle towards the center sites of the defect (**Fig. 6B, 7B**). We have added relevant descriptions in discussion section, please refer to the revised manuscript.

Comment #8. The study does not document the general welfare of the animals, along with appearance of injection site and locomotive behavior of the animal. That can also add some context on visible swelling, redness, edema and abnormal color of the injection sites.

Response: Thank you very much for your reminding. We have supplemented the general welfare of the animals in the materials and methods section (line 823-836). In addition, we recorded the local reaction of injection site after injecting different materials in rat subcutaneous injection model and VML model, including visible swelling, redness, edema and abnormal color, as shown in **Fig. R16, R17**. After subcutaneously injecting different materials (**Fig. R16**), the injection site in all groups showed slight redness and swelling. In addition, the skin protrusion in the mECM and mECM+PM groups was higher than that in the PM group, which was mainly due to the physical support of the injected materials. However, the skin protrusions immediately disappeared in the PM group, because the PM was prone to diffuse to surrounding tissues at the injection site. In the following 28 days, there was no visible redness or swelling at the injection site in all groups until the hair restored to normal state. Moreover, all rats moved normally during the observation period.

Fig. R16. General observation of subcutaneous injection sites of different materials at different time points in rat subcutaneous injection model.

Similarly, we also recorded the local appearance of surgical sites implanted with different materials (**Fig. R17**). The surgical sites in all groups showed slight redness and swelling 4 days after implantation, and gradually subsided on days 7. During the subsequent observation period (7-112 days), rat hairs grew and covered the surgical areas, without any signs of suppuration, edema or swelling. In addition, we also observed the activity behavior of rats after surgery. The walking frequency of the rats significantly decreased 4 days after surgery, but gradually recovered on days 7, and approached normal state on days 28.

Fig. R17. The appearance images of VML treated with different materials at different time points post-operation.

Comment #9. Local tissue response at the interface between the injected biomaterial and the native tissue should be incorporated as well. The number of lymphocytes, polymorphonuclear cells etc should be counted between field of view between interface and implant.

Response: Thanks for your advice, we have evaluated the local tissue response at the interface between the injected biomaterial and the native tissue. The number of foreign body giant cells (FBGCs) was quantified with H&E staining images in **Fig. R18 (Fig. 6I in revised manuscript)**. At 1 week, the number of FBGC in each group was low,

and showed no significant difference among all groups. At 4 weeks, the number of FBGCs slightly increased in both the PM group and the mECM+PM group, it was significantly higher in the PM group than that in the mECM+PM group. No FBGCs were detectable in the mECM group because of the mECM degradation.

Fig. R18. Statistical results of foreign body giant cells (FBGC) in different groups at 1 and 4 weeks (n=3 samples per group, 2 random fields per sample).

Then, we performed immunofluorescence staining on leukocyte cells (CD45⁺), T cells (CD3⁺) and B cells (CD20⁺) in **Fig. R19 (Fig. S12 in revised manuscript)**. At 1 week, CD45⁺ cells were mainly distributed at the boundary of materials in all groups, and it was significantly higher in the mECM group and mECM+PM group than that in the PM group. A small number of CD45⁺ cells can infiltrate into the interior of materials, and the number of CD45⁺ cells in the mECM group and mECM+PM group was higher than that in the PM group, showing no significant difference. At 4 weeks, the number of CD45⁺ cells significantly decreased, and there was no significant difference between the PM group and the mECM+PM group (**Fig. R19A, D**). Furthermore, only a small number of CD3⁺ T cells and CD20⁺ B cells was distributed in surrounding of the materials at 1 and 4 weeks, and there was no significant difference between two groups (**Fig. R19B-C, E-F**).

Fig. R19. A-C) Immunofluorescence staining images of CD45, CD3 and CD20 in different groups 1 and 4 weeks after subcutaneous injection. Scale bar for low magnification: 100 μm , high magnification: 20 μm . D-F) Statistical analysis results of CD45, CD3 and CD20 (n=3 samples per group, 2 random fields per sample).

Comment #10. Evaluation of the thickness of the fibrotic capsule around the implant should also be included in the study. This can be estimated from the presence and thickness of a tightly packed layer of fibroblasts displaying increased eosin uptake.

Response: We appreciate your suggestions. We didn't observe dense fibrous encapsulation in all groups. Although collagen deposition was observed around the materials, a large number of functional capillaries were also distributed inside the collagen, which was not a typical foreign body reaction (FBR). In fact, we didn't detect immune encapsulation reactions after implantation of different materials at 1 and 4 weeks, and the material boundaries were visible (**Fig.6B**). The FBR of implanted biomaterials will greatly affect their subsequent biological functions. The dense fibrous

layer and collagen encapsulation seriously hinder the integration of the material with surrounding tissues, and limit the diffusion of nutrients, oxygen and metabolites [1]. In addition, vascularization and foreign body giant cells (FBGC) are two important indicators for evaluating the FBR of implanted materials [1-4]. The implanted materials with weak FBR promote vascularization, while the material with strong FBR is more likely to be wrapped by dense avascular collagen layers [1].

We evaluated the collagen deposition and distribution through Masson staining in **Fig. R20 (Fig. 6F-H in revised manuscript)**. At 1 week, loose collagen was observed at the boundary between tissue and materials in the mECM and mECM+PM groups, there was a small amount of collagen distribution around the porous microspheres in the PM group, with a thickness of $23.5 \pm 5.4 \mu\text{m}$. However, a large number of functional capillaries were detected inside collagen areas. The number of capillaries was significantly higher in the mECM group and mECM+PM group than that in the PM group, indicating the mECM could promote capillary formation (**Fig. R20A-C**). At 4 weeks, mECM hydrogel degraded and collagen was distributed among the porous microspheres. The thickness of collagen layers distributed at the boundary of the material and tissue was $91.3 \pm 23.9 \mu\text{m}$ in the PM group, and $213.4 \pm 84.5 \mu\text{m}$ in the mECM+PM group, a large number of capillaries still existed inside the collagen in the PM and mECM+PM group, which was not a foreign body collagen encapsulation reaction (**Fig. R20A-C**). In addition, the number of FBGCs distributed around the material boundary is low in all groups (**Fig. R18**).

In summary, a large number of functional capillaries distributed inside the collagen deposition areas, which can promote substance exchange between surrounding tissues and infiltrated tissue inside the materials, as well as integration of implanted material with surrounding tissues. These confirmed that the composites possess good biocompatibility.

[1] D. Zhang, Q. Chen, C. Shi, J. Wan, R. Liu. Dealing with the foreign-body response to implanted biomaterials: strategies and applications of new materials, *Advanced Functional Materials* 31(6) (2020).

[2] L. Zhang, Z. Cao, T. Bai, L. Carr, B.D. Ratner, S. Jiang. Zwitterionic hydrogels implanted in mice resist the foreign-body reaction, *Nature Biotechnology* 31(6) (2013) 553-556.

[3] D. Dong, C. Tsao, H.-C. Hung, F. Yao, S.G.-h. Tang, P. Jain, S. Jiang. High-strength and fibrous capsule-resistant zwitterionic elastomers, *Science Advances* 7(1) (2021).

[4] P. Pakshir, F. Younesi, K.-A. Wootton, A. Daley, I. Parrag, W. Naimark, B. Hinz. Controlled release of low-molecular weight, polymer-free corticosteroid coatings suppresses fibrotic encapsulation of implanted medical devices, *Biomaterials* 286 (2022).

Fig. R20. Collagen deposition and capillaries formation surround the implants at 1 and 4 weeks. A) Masson staining showing the collagen deposition. Black arrows: capillaries. Scale bar for low magnification: 500 µm, high magnification: 100 µm. B) Statistical result of the thickness of collagen layer (n=3 samples per group, 2 random fields per sample). C) The number of capillaries in A (n=3 samples per group, 2 random fields per sample).

Comment #11. In methodology section where images are used to quantify it should be presented with more details including exclusion and inclusion criteria set for quantification.

Response: Thanks for your reminding. We have supplemented the corresponding description of quantitative methods in statistical analysis section and main text (line 1187-1194). Please refer to the revised manuscript.

Comment #12. Some more cytokines should be included IL-23, IL-1 α , MCP-1, IL-12p70, IL-6, IL-27, IL-17A, IFN- β , and GM-CSF.

Response: According to your suggestion, we have supplemented the cytokines, including IL-1 α , MCP-1, IL-12p70, IL-6 and GM-CSF, as showed in **Fig. 4I-S&S10**. The corresponding description and results were added in revised manuscript.

Comment #13. Line 71.... can result in the strucutal and morphological collapse. “strucutal” spelling mistakes.

line 727.... “in vivo and in vivo” should be corrected.

Response: Sorry for our careless mistake. We have corrected it in line 70 and line 811, please see the revised manuscript.

Reviewers' Comments:

Reviewer #1:

Remarks to the Author:

The author has carefully answered each question as requested by the reviewer and made careful revisions to the article to address my concerns.

Reviewer #2:

Remarks to the Author:

Li et al. provided additional data and sufficiently responded to my comments. They present a versatile composite biomaterial for cytokine delivery; that is injectable and has tunable mechanical properties.

The composite biomaterial itself demonstrates a notable level of bioactivity/therapeutic efficacy, that is minimally improved by the addition of cytokines. While the addition of cytokines is scientifically justified, and the controlled release is characterized, the authors refer to "synergy" between the material and cytokines throughout the manuscript that is not fully supported by the data.

Reviewer #3:

Remarks to the Author:

The authors have made significant efforts to respond appropriately to the reviewer queries.

Response to Reviewers' comments

Reviewer #2 (Remarks to the Author):

Li et al. provided additional data and sufficiently responded to my comments. They present a versatile composite biomaterial for cytokine delivery; that is injectable and has tunable mechanical properties.

The composite biomaterial itself demonstrates a notable level of bioactivity/therapeutic efficacy, that is minimally improved by the addition of cytokines. While the addition of cytokines is scientifically justified, and the controlled release is characterized, the authors refer to “synergy” between the material and cytokines throughout the manuscript that is not fully supported by the data.

Response: Thanks for your in-depth comments. Indeed, the therapeutic efficacy of materials was enhanced by cytokines to a certain extent, while it lacked sufficient data to confirm the synergistic effect between the materials and cytokines. Thus, we have modified the inappropriate description throughout the manuscript.